# Benchmarking metabolic RNA labeling techniques for high-throughput single-cell RNA sequencing

Xiaowen Zhang[1,2,3,6], Mingjian Peng[1,2,3,6], Jianghao Zhu[1,2,3], Xue Zhai[1,2,3], Chaoguang Wei[1,2,3], He Jiao[1,2,3], Zhichao Wu[1,2,3], Songqian Huang[1,2,3], Mingli Liu[1,2,3], Wenhao Li[1,2,3], Wenyi Yang[1,2,3], Kai Miao ®[4], Qiongqiong Xu[1,2,3], Liangbiao Chen ®[1,2,3] & Peng Hu ®[1,2,3,5] ✉

Metabolic RNA labeling with high-throughput single-cell RNA sequencing (scRNA-seq) enables precise measurement of gene expression dynamics in complex biological processes, such as cell state transitions and embryogenesis. This technique, which tags newly synthesized RNA for detection through induced base conversions, relies on conversion efficiency, RNA integrity, and transcript recovery. These factors are influenced by the chosen chemical conversion method and platform compatibility. Despite its potential, a comprehensive comparison of chemical methods and platform compatibility has been lacking. Here, we benchmark ten chemical conversion methods using the Drop-seq platform, analyzing 52,529 cells. We find that on-beads methods, particularly the meta-chloroperoxy-benzoic acid/2,2,2-trifluoroethylamine combination, outperform in-situ approaches. To assess in vivo applications, we apply these optimized methods to 9883 zebrafish embryonic cells during the maternal-to-zygotic transition, identifying and experimentally validating zygotically activated transcripts, which enhanced zygotic gene detection capabilities. Additionally, we evaluate two commercial platforms with higher capture efficiency and find that on-beads iodoacetamide chemistry is the most effective. Our results provide critical guidance for selecting optimal chemical methods and scRNA-seq platforms, advancing the study of RNA dynamics in complex biological systems.

Single-cell RNA sequencing (scRNA-seq) has revolutionized our understanding of cellular heterogeneity and transcriptomic complexity. However, traditional scRNA-seq methods often fail to capture the temporal dynamics of RNA. Recent advances in time-resolved high-throughput scRNA-seq using metabolic labeling have provided deeper insights into RNA dynamics in complex biological processes[1–8].

In metabolic labeling assays, nucleoside analogs, such as 4-Thiouridine (4sU)[9–15], 5-Ethynyluridine (5EU)[16–20], and 6-Thioguanosine (6sG)[1,21–25], are rapidly incorporated into newly synthesized RNA, creating a chemical tag that can be detected via sequencing by identifying chemical-induced conversions. This strategy is applicable to a wide range of model organisms, including humans[5,26], mice[2,6], zebrafish[7,8,27],

[1]Key Laboratory of Exploration and Utilization of Aquatic Genetic Resources, Ministry of Education, Shanghai Ocean University, Shanghai, China. [2]International Research Center for Marine Biosciences, Ministry of Science and Technology, Shanghai Ocean University, Shanghai, China. [3]Center for Aquacultural Breeding Research, Shanghai Ocean University, Shanghai, China. [4]MOE Frontier Science Centre for Precision Oncology, University of Macau, Macau SAR, China. [5]Marine Biomedical Science and Technology Innovation Platform of Lin-gang Special Area, Shanghai, China. [6]These authors contributed equally: Xiaowen Zhang, Mingjian Peng. ✉e-mail: phu@shou.edu.cn

and fruit flies[14]. Metabolic RNA labeling combined with scRNA-seq has significantly enhanced our ability to quantitatively analyze RNA synthesis and degradation. This approach has enabled key discoveries, such as understanding cell-cycle dynamics in cultured cells and organoids[3,4,16,26], tracking RNA during embryogenesis[7,8,27,28], investigating transcriptional bursting[29], and identifying rapid transcriptional responses during viral infection[1].

Metabolic labeling involves several crucial steps: incorporating 4sU into newly synthesized RNA, performing chemical conversion reactions, and ensuring compatibility with high-throughput scRNA-seq platforms. Key chemical conversion methods include SLAM-seq[1,9], which uses an iodoacetamide (IAA)-based reaction; TimeLapse-seq[2,10], which utilizes 2,2,2-trifluoroethylamine (TFEA) with meta-chloroperoxy-benzoic acid (mCPBA)/sodium periodate (NaIO$_4$) and NH$_4$Cl-based reactions; and TUC-seq[30], which involves osmium tetroxide (OsO$_4$) and ammonium chloride (NH$_4$Cl). These steps determine the assay's efficiency, including conversion efficiencies as indicated by T-to-C substitution rates, and RNA recovery rates, indicated by the number of genes and transcripts detected per cell. Although statistical methods have been developed to correct some of these limitations[2,10,12], improving 4sU labeling and T-to-C conversion efficiency at the experimental level remains crucial for achieving more reliable and consistent outcomes, advancing our understanding of RNA dynamics in various biological processes.

Several methods have been developed to integrate metabolic labeling with high-throughput scRNA-seq platforms, such as scNT-seq[2], scSLAM-seq[1,7], sci-fate[3], sci-fate2[4], 10× Genomics-based method[8], and Well-TEMP-seq[5,6]. These methods are built upon different scRNA-seq platforms, each with unique technical adaptations. For example, scNT-seq[2] is based on the Drop-seq platform[31] and leverages a microfluidic device, a strategy also implemented in the commercial 10× Genomics system[32]. Well-TEMP-seq[5,6] employs a microwell-based system, while sci-fate and sci-fate2 utilize the sci-RNA-seq approach[3,4,33], which relies on multiple rounds of split-pool barcoding. The key distinction among these methods is the timing of chemical conversion, which occur either before or after single-cell encapsulation, potentially affecting conversion rates. scNT-seq[2] relies on the home-brew Drop-seq platform[31], while Well-TEMP-seq[5,6] relies on the Well-Paired-seq platform[34], both of which use the same barcoded beads from Drop-seq and enable chemical conversion on naked capture RNA attached to barcoded beads after cell lysis. In contrast, sci-fate, sci-fate2 and 10× Genomics-based methods employ in-situ IAA-based chemical conversion within cells before mRNA releasing from the intact cell (Supplementary Fig. 1). Compared to the relatively low cell capture rate of the home-brew Drop-seq platform (~5%)[31], in-situ IAA chemical conversion coupled with commercial platforms, such as 10× Genomics[32] and MGI C4[35], with higher capture rates (~50%)[32] can be more effective for studying unique biological systems, such as early-stage embryos, where only a limited number of cells are available[8]. Although these methods have been demonstrated in different cell lines and biological systems, they vary in conversion efficiency, RNA recovery rate, and compatibility with scRNA-seq platforms[2,3]. Given these differences and their potential impact on RNA dynamics analysis, a systematic and unbiased comparison of chemical conversion methods and their compatibility with different single-cell platforms is needed.

To address this gap, we tested ten chemical conversion methods with varying reagent components and buffer conditions (Fig. 1a, b), including comparisons of in-situ and on-beads conditions using the same cell line. Our work provides direct comparisons and recommendations for time-resolved scRNA-seq methods using metabolic labeling. We further demonstrated that the recommended method effectively identifies zygotically activated transcripts in zebrafish embryogenesis. Additionally, we compared the 10× Genomics and MGI C4, two commercial single-cell platforms with high cell capture efficiency to the home-brew Drop-seq platform. The results highlight the strengths and weaknesses of each system, offering guidance for selecting the most appropriate chemical reaction and single-cell platform.

## Results

### Experimental design and computational quality control assessment

To ensure a comprehensive benchmarking, we summarized currently available metabolic labeling scRNA-seq methods (Supplementary Fig. 1). These methods are built upon different scRNA-seq platforms, each incorporating unique technical adaptations in chemical conditions and the timing of the chemical conversion step. Given this variability, we focused our direct comparison on the Drop-seq platform, which utilizes barcoded beads specifically designed to capture polyA-tailed mRNA directly onto the beads. This setup allows for buffer exchange and enables on-beads chemical conversion reactions prior to reverse transcription, making the platform widely adopted and customizable[31] (Fig. 1a). We investigated two widely used chemical approaches: SLAM-seq[1,9], which utilizes an iodoacetamide (IAA)-based reaction, TimeLapse-seq[2,10], which employs 2,2,2-trifluoroethylamine (TFEA) in combination with oxidizing agents meta-chloroperoxy-benzoic acid (mCPBA) or sodium periodate (NaIO$_4$). Additionally, we included NH$_4$Cl-based reactions adapted from TUC-seq[30] (Fig. 1b and Supplementary Fig. 1). To optimize conditions, we varied the pH or temperature conditions of these reactions based on recent metabolic labeling bulk RNA-seq studies[36]. To further explore the impact of timing in chemical conversion steps, we performed in-situ chemical conversion using SLAM-seq or 10× Genomics-based method as previously described[3,8]. In total, we conducted ten chemical conversion comparisons using the ZF4 fibroblast cell line derived from zebrafish embryos (Fig. 1a, b). To ensure consistency in cell processing, ZF4 cells were fixed with methanol after metabolic labeling (100 μM 4sU) for 4 h. Chemical conversion was performed either in situ before single-cell encapsulation or on beads after encapsulation. Libraries from different methods were prepared and sequenced.

For data analysis, we used the dynast pipeline[37] and developed a dedicated pipeline for quality control (untreated) and method comparison (see "Methods" and Fig. 1c). We evaluated and compared chemical conversion methods based on three criteria: (1) RNA integrity (cDNA size), (2) conversion efficiency (T-to-C substitution rate), and (3) RNA recovery rate (number of genes and unique molecular identifiers (UMIs) detected per cell) (Fig. 1d).

After quality filtering, we obtained 22,955 single-cell transcriptomes for ZF4 cells, with a median of 2472 UMIs corresponding to transcripts and 1109 genes detected per cell. Analysis of base mutations in each mapped transcript revealed a significant increase in T-to-C substitution rates in all chemically treated samples compared to the control condition without chemical conversion treatment, while other conversion rates remained below or around background levels (Supplementary Fig. 2a). The top three chemical conversion methods—mCPBA/TFEA pH 7.4, mCPBA/TFEA pH 5.2, and NaIO$_4$/TFEA pH 5.2—had average T-to-C substitution rates of 8.40%, 8.11%, and 8.19%, respectively (Fig. 2a and Supplementary Table 1). Additionally, more than 40% of mRNA UMIs were labeled per cell (Supplementary Fig. 2b), demonstrating the protocols' efficiency with fixed/cryo-preserved cells. Notably, one condition, on-beads IAA at 37 °C, exhibited a relatively low T-to-C substitution rate of 3.84% but an unexpectedly high proportion of labeled mRNA UMIs per cell (45.98%) (Fig. 2a and Supplementary Fig. 2b). Upon analyzing the mutation frequency and distribution, we found that this condition tends to label a broader range of RNA molecules rather than introducing multiple substitutions within the same RNA strand (Supplementary Fig. 2c).

We also compared the same chemistry condition, which is conducted in intact cells (in-situ) to after mRNA release from the cells and attached to the beads (on-beads). The on-beads method achieved a

**a**  **Experimental workflow**

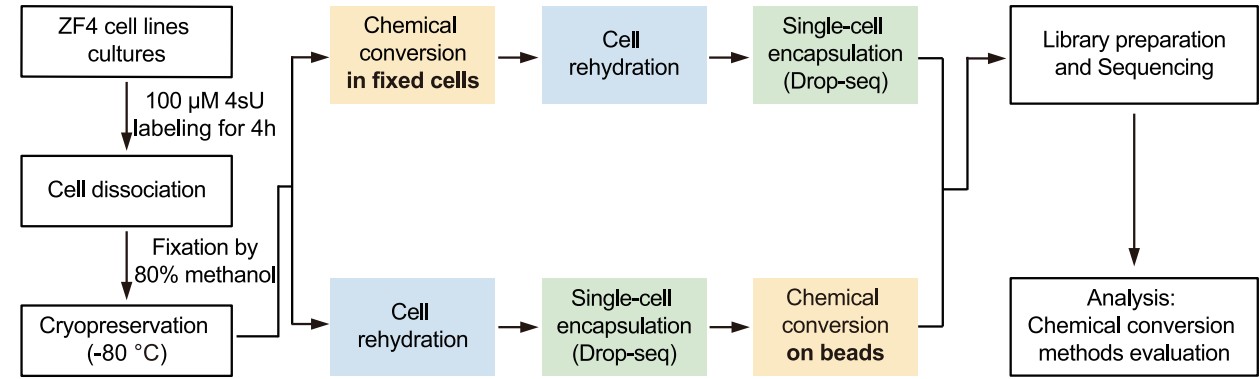

**b**  **Chemical conversion methods**

| Method | | Reagent | Buffer pH | Temperature | Time | Reference |
|---|---|---|---|---|---|---|
| SLAM -based | In-situ (In fixed cells) | IAA | 7.4 | 50 °C | 15 mins | *Herzog et al.*, 2017 (SLAM-seq); *Holler et al.*, 2021. |
| | In-situ (In fixed cells) | IAA | 8.0 | 50 °C | 15 mins | *Herzog et al.*, 2017 (SLAM-seq); *Cao et al.*, 2020 (Sci-fate); *Maizels et al.*, 2024 (Sci-fate2). |
| | On-beads | IAA | 8.0 | 32 °C | 15 mins | *Fishman et al.*, 2024 (scSLAM-seq). |
| | On-beads | IAA | 8.0 | 37 °C | ~1 hr | *Lin et al.*, 2023 (Well-TEMP-seq); *Yin et al.*, 2024 (Well-TEMP-seq). |
| TimeLapse -based | On-beads | mCPBA+TFEA | 5.2 | 45 °C | ~1.5 hrs | *Schofield et al.*, 2018 (TimeLapse-seq); *Zimmer et al.*, 2021 (STL-seq); *Zimmer et al.*, 2023. |
| | | mCPBA+TFEA | 7.4 | 45 °C | ~1.5 hrs | *Zimmer et al.*, 2023. |
| | | NaIO$_4$+TFEA | 5.2 | 45 °C | ~1.5 hrs | *Schofield et al.*, 2018 (TimeLapse-seq); *Qiu et al.*, 2020 (scNT-seq); *Zimmer et al.*, 2023. |
| | | NaIO$_4$+TFEA | 7.4 | 45 °C | ~1.5 hrs | *Zimmer et al.*, 2023. |
| | | NaIO$_4$+NH$_4$Cl | 8.8 | 45 °C | ~1.5 hrs | *Zimmer et al.*, 2023. |
| TUC-based | On-beads | OsO$_4$+NH$_4$Cl | 8.8 | 50 °C | ~3 hrs | *Riml et al.*, 2017 (TUC-seq); *Zimmer et al.*, 2023. |

**c**  **Computational pipeline**

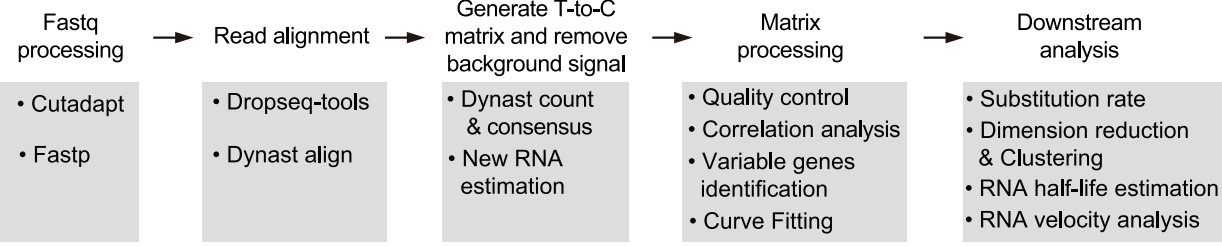

**d**  **Chemical conversion methods evaluation**

| I. cDNA size | II. T-to-C substitution rate | III. Number of genes or UMIs detected per cell |

2.32-fold higher substitution rate than the in-situ method (mean of 6.07% versus 2.62%). The T-to-C substitution rate of on-beads IAA (32 °C) was also comparable to the top three methods, with a median of 6.57% and a mean of 6.39% (Fig. 2a and Supplementary Table 1), resulting in an average of 36.87% of total mRNAs being labeled (Supplementary Fig. 2b). The sensitivity of gene detection is crucial for scRNA-seq performance, and all chemical conversion treatments compromised library complexity to some extent, consistent with previous findings[2,38]. Among the chemical conversion methods, the mCPBA/TFEA pH 5.2 reaction outperformed the others, minimally impacting cDNA size integrity (Supplementary Fig. 2d, e) and detecting approximately 2044 genes and 5468 UMIs per cell at a sequencing depth of 10,000 reads per cell, comparable to untreated samples (Fig. 2b, c and Supplementary Table 1).

**Fig. 1 | Experimental design for benchmarking chemical conversion methods.**
**a** Workflow for high-throughput scRNA-seq using metabolic labeling in ZF4 cells. ZF4 cells were labeled with 4-thiouridine (4sU, 100 μM), followed by cell dissociation and fixation. Chemical conversion was performed either before or after single-cell encapsulation on the Drop-seq platform. Newly synthesized transcripts were detected via sequencing by identifying chemical-induced T-to-C substitutions.
**b** Summary of the ten chemical conversion methods evaluated in this study, including key parameters such as the main reagent, buffer pH, temperature, reaction time, and relevant references. "In-situ" refers to chemical conversion occurring within intactly fixed cells, while "on-beads" indicates that the chemical conversion occurs after mRNA is released from the cells and captured on beads. IAA

iodoacetamide, mCPBA meta-chloroperoxy-benzoic acid, TFEA 2,2,2-tri-fluoroethylamine, $NaIO_4$ sodium periodate, $NH_4Cl$ ammonium chloride, $OsO_4$ osmium tetroxide. **c** Computational pipeline for data processing, starting with fastq file pre-processing using Cutadapt and fastp, followed by read alignment with Dynast and Dropseq-tools. T-to-C substitutions were identified using Dynast, with R and Python scripts used for cell quality control, dimension reduction, new transcript identification, and RNA velocity analysis (see details in "Methods"). **d** Benchmarking criteria used to evaluate chemical conversion performance, focusing on cDNA size, T-to-C substitution rate, and the number of genes and unique molecular identifiers (UMIs) detected per cell.

Collectively, these data provide a direct comparison and benchmarking of various widely adopted chemical conversion methods.

## mRNA control strategy during the cell cycle

Applying metabolic RNA labeling to high-throughput single-cell assays enables the measurement of temporal gene expression dynamics, distinguishing between newly synthesized and pre-existing RNA during processes such as cell state transitions. After quality filtering, we obtained 52,529 cells, which were classified into two clusters: steady-state and dividing cells, based on the relatively high expression of well-known mitotic cell cycle genes (Fig. 2d and Supplementary Fig. 3a). Cell type composition varied across chemical conversion conditions, with $OsO_4$ and IAA (37 °C) treatments depleting dividing cells (Supplementary Fig. 3b, c). This suggests these conditions may impair the detection of proliferative cells by affecting RNA stability or conversion efficiency. To ensure that 4sU labeling itself does not alter cell cycle dynamics, we performed flow cytometry analysis, which confirmed that the addition of 4sU did not affect cell cycle distribution (Supplementary Fig. 3d). Pearson correlation analysis revealed high concordance between the on-beads methods for labeled and unlabeled RNAs (Pearson's $r > 0.9$), whereas the correlation was lower (Pearson's $r$: 0.50–0.88) between in-situ and any of the on-beads methods. This indicates that performing chemical conversion in intact cells versus after mRNA release introduces more transcriptomic variation than the choice of chemical method alone (Supplementary Fig. 4a).

Since marker genes in dividing cluster are enriched in mRNA processing (Supplementary Fig. 4b), indicating active RNA synthesis, we benchmarked the chemical conversion methods for their ability to identify genes involved in the cell cycle. The expression of genes involved in cell cycles, including *tubb4b* and *ccnd1*, showed a higher proportion of labeled transcripts in single cells, while the housekeeping genes like *rplp0*, known for their stability, maintained consistent expression across all conditions (Fig. 2e, Supplementary Fig. 5a, b, and Supplementary Table 2). These results further indicate that a higher T-to-C substitution rate correlates with an increased proportion of newly synthesized, labeled transcripts (Fig. 2e). Among the labeled transcripts, mCPBA-based chemical conversion methods demonstrated the highest conversion efficiency, without compromising library complexity in terms of the number of transcripts identified, compared to the other chemical conversion methods (Fig. 2e).

To further investigate RNA turnover in steady-state cells, we estimated the mRNA half-life based on the proportion of labeled transcripts for each gene (see "Methods"). We analyzed the half-lives of 17,653 detected genes to assess mRNA stability (Supplementary Fig. 6). As expected, cell cycle genes exhibited a faster turnover rate and shorter half-lives compared to housekeeping genes (Supplementary Fig. 6). Additionally, the top 10% most stable and unstable transcripts were enriched for GO terms similar to those found in mouse embryonic stem cells[2] (Supplementary Fig. 6). Unstable transcripts were primarily associated with cell cycle and transcription regulation, while stable transcripts were linked to oxidative phosphorylation, highlighting the conserved roles of transcription and oxidative phosphorylation between mammals and fish.

## Identification of zygotically activated transcripts in zebrafish embryogenesis

To evaluate the performance of optimal chemical conversion method in vivo, we focused on the maternal-to-zygotic transition in zebrafish. Zebrafish embryos were injected at the one-cell stage with 4sU, and cells were harvested at specific time points post-fertilization to identify newly transcribed, zygotically activated transcripts and distinguish them from maternally deposited RNAs (Fig. 3a). Since the proportion of zygotically activated transcripts is low, accounting for less than 1% before 4 h post-fertilization (hpf), we focused on the 5.5 hpf stage, where zygotic transcripts represent 9.33% of the total RNA pool[27]. After quality filtering, we analyzed 9883 embryonic cells, classifying them into six clusters based on previously reported markers[7,8,39] (Fig. 3b, c).

To quantitatively distinguish the fractions of maternal and zygotic transcripts for each gene, we estimated newly transcribed mRNA by modeling from labeled transcripts, enabling us to deduce zygotic mRNA from the total pool. We further partitioned expressed genes into 10 equally sized bins (quantiles) based on their fraction of newly transcribed zygotic mRNA, as previously described[7] (see "Methods" and Supplementary Fig. 7a). We benchmarked our ability to identify zygotic genes based on the proportion of new RNA, comparing our results to two published studies: one on 5.3 hpf embryos using on-beads IAA-based method[7], and the other on 6 hpf embryos using in-situ IAA-based method[8]. For a direct comparison, we identified 14,043 commonly expressed genes across our three datasets and the two public datasets. We defined zygotic genes based on their higher proportions of newly transcribed RNA, using different new-to-total RNA ratio (NTR) cut-offs (70%, 75%, 80%, and 85%; see "Methods" and Supplementary Fig. 7a). This approach allowed us to classify the 14,043 commonly expressed genes into three categories: maternal-only (M), zygotic-only (Z), and both maternally contributed and zygotically expressed (MZ).

We compared the number of genes in each category (M, Z, and MZ) across our datasets and the public datasets, finding that our mCPBA-based methods identified a higher number and proportion of zygotically expressed genes (Fig. 3d, e and Supplementary Fig. 7b, c), likely due to the higher T-to-C substitution rates observed (Supplementary Fig. 7d). Approximately 78% of genes with newly transcribed RNA were classified as MZ genes (Fig. 3e), indicating that these genes were maternally provided but also newly transcribed during zygotic genome activation (ZGA). This proportion is highly consistent with previous reports from bulk RNA-seq in zebrafish (74% MZ genes)[40] and nascent RNA-seq in *Xenopus* (78.4% MZ genes)[41]. Notably, as the NTR ratio used to define zygotic gene classification increased, the difference in the number of identified zygotic genes between methods with higher T-to-C substitution rates and those with lower rates also increased (Fig. 3d and Supplementary Fig. 7b–d). This highlights the significant advantage of optimizing the chemical conversion step for improving zygotic gene identification. For example, using a 70% NTR cut-off, 452 genes were shared across all five datasets, including *tbx16*, *marcksl1b*, and *cited4b*, while 458 zygotic genes were identified across the four on-beads methods (e.g., *apoeb*), and 131 zygotic genes were

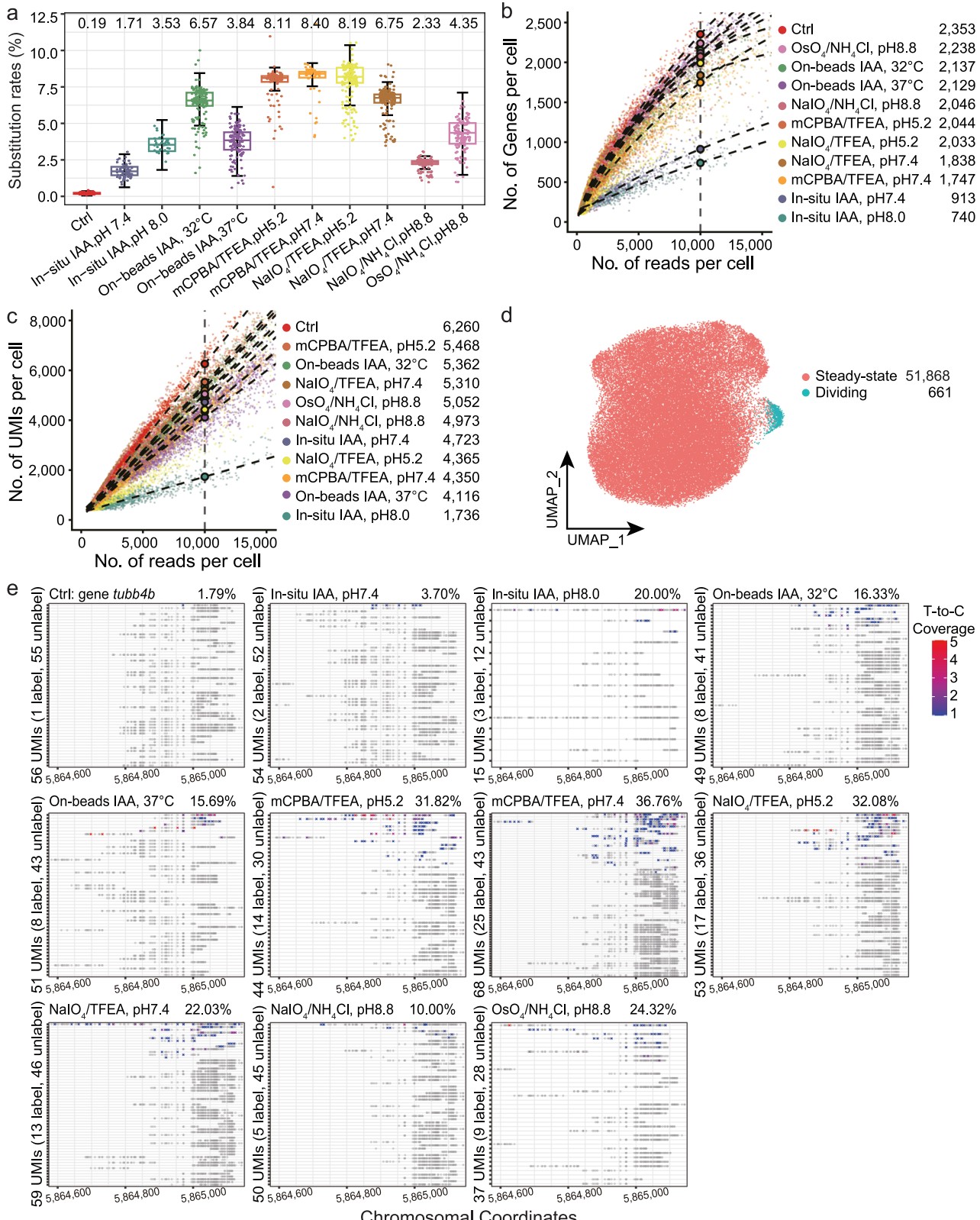

shared between our datasets and the in-situ method (e.g., *akap12b*) (Fig. 3f and Supplementary Fig. 8a, b). We additionally identified 380 zygotic genes using mCPBA/TFEA method with a higher T-to-C substitution rate (pH 7.4), such as *pnrc2* (Fig. 3f). Overall, the on-beads methods consistently demonstrated higher conversion rates than the in-situ method, resulting in a greater proportion of labeled mRNA per cell (Supplementary Fig. 7d).

Whole-embryo in situ hybridization confirmed the expression of selected representative genes at the animal pole at 5.5 hpf, validating the accuracy of our classification and demonstrating the effectiveness of our methods in identifying zygotic transcripts during zebrafish embryogenesis (Fig. 3g). To confirm that these transcripts were newly synthesized, we designed intronic probes for six genes and additionally included *marcksa* and *setdb1b*, which were exclusively identified

**Fig. 2 | Comparison and evaluation of ten chemical conversion methods using ZF4 cells. a** Box plot showing T-to-C substitution rates across control and ten chemical conversion methods in 4sU-labeled ZF4 cells. "Ctrl" denotes the untreated control group (*n* = 7531 cells). The ten chemically treated groups, from left to right, contain: *n* = 1587, 789, 5581, 5267, 4692, 4639, 5360, 6461, 5389, and 5233 cells. Box plots show the median (center line), interquartile range (box), 1.5× interquartile range. Source data are provided as a Source Data file. **b, c** Scatterplots comparing the number of genes (**b**) or UMIs (**c**) detected per cell as a function of aligned reads per cell across the ten chemical conversion methods. Color indicates treatment methods. Fitted lines for each method are included, along with the predicted number of genes or UMIs detected per cell at a sequencing depth of 10,000 reads. In (**b**), the curve is smoothed using locally weighted regression (LOESS), while in (**c**), a linear model (LM) is applied. The estimated numbers of genes or UMIs at 10,000 reads are displayed on the right of the figure. Source data are provided as a Source Data file. **d** Uniform Manifold Approximation and Projection (UMAP) visualization showing integrated control and datasets from the ten chemical conversion methods, representing 52,529 ZF4 cells. Cells are colored by cell type. The numbers of cells in each group are also indicated. **e** Visualization of unique transcripts (UMIs) of the cell-cycle gene *tubb4b* from individual ZF4 cells in the control group and across the ten chemical conversion methods. Grey circles represent uridines without T-to-C substitution, while crosses ("X"s) indicate uridines with T-to-C substitutions in at least one read. The read coverage for each T-to-C substitution is displayed with color scaling.

using the mCPBA/TFEA method. Their unspliced transcript expression at 5.5 hpf was validated using control probes for comparison (Supplementary Fig. 8c). Additionally, we compared our dataset with a zygote-specific gene list derived from intron signal capture in bulk assays[40]. Our 4sU-labeled zygotic genes showed a strong overlap with this dataset (odds ratio = 12.13, *p* < 2.2e-16, Fisher's exact test, Supplementary Fig. 9a), further supporting the robustness of our approach in identifying zygotic gene activation. Furthermore, RNA velocity analysis based on metabolic labeling accurately captured the temporal dynamics of gene expression in single cells during development, from a proliferating progenitor state (primordial germ cell, PGC) to a more differentiated state, such as enveloping layer (EVL) (Supplementary Fig. 9b). This result aligns with previous findings[2,37], showing that metabolic labeling methods outperform splicing-based RNA velocity predictions in accurately predicting cell fate. Moreover, cell cycle scoring analysis revealed that PGCs had relatively fewer cells in the G1 phase, exhibited lower zygotic transcription, and retained maternal mRNAs (Supplementary Fig. 9c, d). This aligns with the progressive addition of the G1 phase during ZGA[42].

While cell cycle arrest can enhance zygotic transcription, its influence on cell-type-specific gene activation and the timing of ZGA remains under investigation. In frogs, cell cycle lengthening promotes ZGA[41], but in zebrafish, Chk1 overexpression slows cell division without affecting ZGA[42]. Leveraging our dataset, we found that ~38.8% of Chk1 overexpression-induced genes overlapped with zygotic genes, suggesting that while ZGA timing is independent of cell cycle length, an extended cycle enhances transcriptional competence (Supplementary Fig. 9e, f). Moreover, Chk1-induced zygotic genes showed cell-type-specific expression, with *grhl3* and *cldne* enriched in the EVL and *lft2* and *tbxta* in endoderm and mesoderm clusters (Supplementary Fig. 9g). These findings indicate that cell cycle lengthening primes lineage-specific gene activation, shaping early developmental transcriptional programs.

## Comparison between different high-throughput single-cell platforms

The cell capture efficiency of the Drop-seq platform is around 5%[31], which is relatively low and may be challenging when working with samples that have limited cell numbers, particularly during early embryo development. To address this limitation, we evaluated two widely used commercial microfluidic droplet-based high-throughput platforms: 10× Genomics[32,43] and MGI C4[35] platform. Both platforms offer significantly higher capture efficiencies (~50%) but differ in their flexibility for chemical conversion steps. The MGI C4 platform enables on-beads chemical conversion, similar to Drop-seq, whereas the commercial 10× Genomics platform completes reverse transcription with single-cell co-encapsulation, necessitating in-situ chemical conversion for metabolic labeling (Fig. 4a, b).

We tested various chemical conversion methods on the MGI C4 platform using hard magnetic barcoded beads and found that TFEA-based methods were incompatible with these beads, leading to low sequencing library complexity. Given that on-beads IAA (32 °C) method showed a comparable T-to-C substitution rate to

the highest mCPBA-based method (Fig. 2a), we focused on this method.

To comprehensively evaluate chemical conversion strategies, we included all possible method-platform combinations: in-situ chemical conversion (in-situ IAA, pH8.0) with 10× Genomics, C4, and Drop-seq, as well as on-beads conversion (On-beads IAA, 32 °C) with C4 and Drop-seq. The results showed: (1) For the same in-situ chemistry, C4 exhibited the highest T-to-C substitution rate (5.74%) (Fig. 4c, d). Both 10× Genomics (nGene: 670, nUMI: 1745) and C4 (nGene: 624, nUMI: 1263) showed comparable library complexity at 4000 reads per cell, with both outperforming the customized Drop-seq platform, as expected (Fig. 4e, f). Notably, the number of detected genes using in-situ chemistry with 10× Genomics aligns with previous studies (nGene: 626, nUMI: 923) (Supplementary Fig. 10a, b), further validating the robustness of our findings; (2) For the same platform, on-beads chemical reactions outperformed in-situ reactions in terms of T-to-C substitution rate (Drop-seq: 6.57% vs. 3.53%; C4: 8.44% vs. 5.74%) (Fig. 4d); (3) Overall, on-beads chemical conversion with the MGI C4 platform outperformed all other methods, achieving the highest T-to-C mutation rates and superior library complexity (Fig. 4e, f).

In summary, the MGI C4 platform, which supports on-beads chemical conversion, offers the highest sensitivity with the highest T-to-C substitution rate (8.44%) and superior transcript detection, making it ideal for rare cell populations such as early embryos. Meanwhile, 10× Genomics, a widely adopted platform, is restricted to in-situ chemical conversion, which reduces conversion efficiency but maintains the highest library complexity. In contrast, the home-brew Drop-seq system, while flexible, suffers from lower cell capture efficiency (~5%) and reduced library complexity, making it optimal for limited input cell number (Supplementary Fig. 11). Cost is also a key consideration when choosing a metabolic labeling scRNA-seq method, especially for large-scale studies or rare cell populations where maximizing transcript recovery is critical (Supplementary Table 3).

## Discussion
Metabolic labeling of artificial nucleosides, such as 4-Thiouridine (4sU)[1,3,9–15,25,44], 5-Ethynyluridine (5EU)[16–20], and 6-Thioguanosine (6sG)[1,21–25,43], enables the simultaneous identification of newly synthesized and pre-existing RNAs in single cells by inducing conversions detectable through sequencing, among which 4sU is the most widely used. This technique integrates efficiently with high-throughput single-cell RNA-seq platforms, enabling accurate, large-scale analysis of RNA dynamics. While metabolic labeling has been applied to various cultured cells using different chemical conversion methods, comprehensive comparisons of conversion efficiency, reverse transcriptase read-through potential, gene detection bias, and library complexity across different 4sU chemical conversion methods have been lacking. In this study, we directly compared ten widely used methods to benchmark their performance, using the same cell line, single-cell platform, and standardized computational preprocessing steps. Additionally, we developed computational pipelines for rigorous quality control (Fig. 1). Our benchmarking results provide essential guidelines for selecting optimal chemical conversion method and

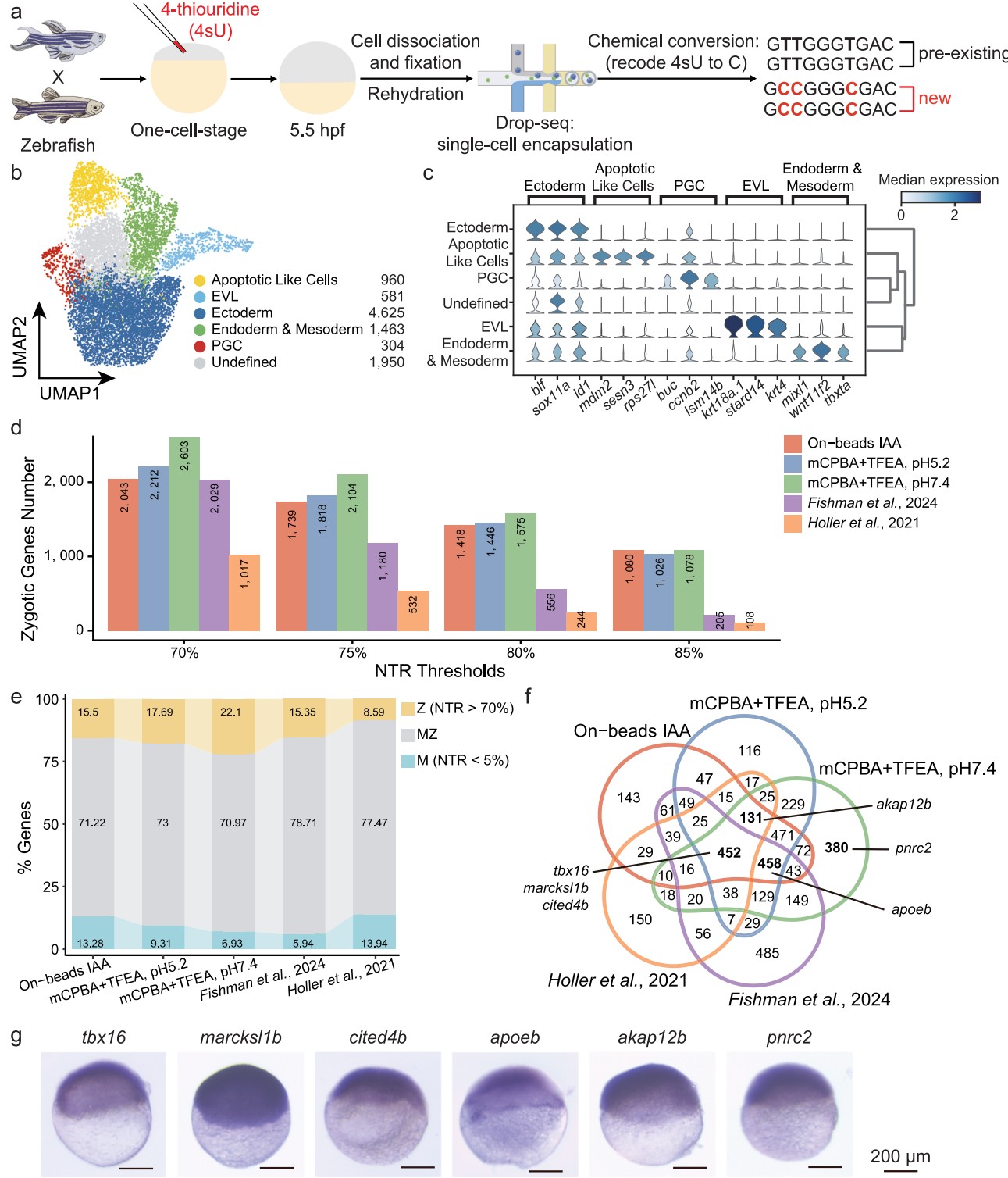

compatible platforms for studying RNA dynamics in complex biological processes in terms of cost, cell type preservation, and sensitivity (Supplementary Fig. 11).

Previous studies have shown that chemical conversion can significantly hinder reverse transcription, potentially reducing library complexity in scRNA-seq by causing incomplete cDNA synthesis[2,38]. We analyzed the cDNA size of libraries generated with and without the ten chemical conversion methods. Consistent with previous findings[2,38], we observed a negative correlation between conversion efficiency and library complexity. Specifically, mCPBA/TFEA-based methods (pH 5.2 and pH 7.4) with second-strand synthesis effectively preserved RNA

integrity, resulting in an average cDNA size comparable to untreated samples while achieving higher conversion efficiency (1091 bp vs. 1103 bp, Supplementary Fig. 2d, e). This method is particularly useful for analyzing chemical conversion rates beyond the 3′ UTR and for accurately predicting cell status transitions during cell cycle regulation, as well as other complex biological systems.

In metabolic labeling experiments, T-to-C substitutions are key indicators of newly synthesized RNA, essential for accurately quantifying RNA dynamics. However, current labeling strategies often result in incomplete 4sU incorporation into newly transcribed RNAs in single cells[2,38]. Although binomial mixture model-based statistical

**Fig. 3 | Identification of zygotically activated transcripts in zebrafish embryogenesis using improved chemical conversion methods. a** Zebrafish embryos were injected at the one-cell stage with 4-thiouridine (4sU, red), which incorporates into newly transcribed zygotic mRNA, leaving pre-existing maternal mRNA unlabeled. Embryos were collected at 5.5 h post-fertilization (hpf), dissociated into single cells, and analyzed using the Drop-seq platform with improved chemical conversion methods, inducing T-to-C substitutions in newly transcribed (zygotic) mRNA. **b** Uniform Manifold Approximation and Projection (UMAP) projection of 9883 single cells from zebrafish embryos at 5.5 hpf, colored by six cell-type clusters. The number of cells in each group is indicated. EVL enveloping layer, PGC primordial germ cell. **c** Violin plot displaying the marker genes for identified cell clusters. The expression level of each marker gene is color-coded based on the median expression in each cluster, with the color gradient ranging from light blue (low expression) to dark blue (high expression), scaled across all clusters. **d** Histogram depicting the number of identified zygotic genes across three chemical conversion methods in our study, compared to published data[7,8]. The x-axis

represents different new-to-total RNA ratio (NTR) thresholds, while the y-axis and the numbers within the bars indicate the gene counts. Source data are provided as a Source Data file. **e** Stacked bar chart showing the proportions of identified maternal (M), maternal-zygotic (MZ), and zygotic (Z) genes (NTR > 70%) across three chemical conversion methods in our study compared to published data[7,8]. Colors indicate gene types. **f** Venn diagram showing the overlap of defined zygotic genes from (**e**) among different chemical conversion methods and published studies, highlighting both unique and shared genes. *Tbx16*, *marcksl1b*, and *cited4b* are identified in all datasets. *Apoeb* is uniquely identified in the on-beads methods across four datasets, excluding the in-situ chemical conversion study by ref. 8. *Akap12b* is detected in four datasets, excluding the study by ref. 7. *Pnrc2* is exclusively detected in our mCPBA/TFEA method (pH 7.4). **g** In-situ hybridization staining validation of 5.5 hpf zebrafish embryos for mRNAs of zygotic genes indicated in (**f**). Scale bar: 200 µm. Each staining pattern was visualized in three independent samples and yielded similar results.

corrections, as implemented in the new RNA estimation[2,10,12] and the dynast software used in this study, can partially mitigate this issue, enhancing conversion efficiency remains critical for achieving higher T-to-C substitution rates. Our study demonstrated that mCPBA/TFEA-based methods (pH 5.2 and pH 7.4) used with the home-brew Drop-seq platform outperformed other methods, achieving a T-to-C substitution rate of > 8% with ~40% transcripts was labelled in adherent cultured ZF4 cells. We note that chemical conditions can affect cell type composition; specifically, IAA chemistry at 37 °C for 1 h, but not at 32 °C for 15 min, introduced a bias by selectively depleting proliferating cell clusters. Alternatively, the low proportion of dividing cells (~3% in ZF4) may hinder accurate quantification (Supplementary Fig. 3b, c). To ensure a direct comparison across different chemical reactions, we also tested these methods on fixed zebrafish embryonic cells. The results showed that mCPBA/TFEA-based methods maintained their efficiency, achieving T-to-C substitution rates comparable to, or slightly higher than, those observed in published datasets using fresh cells[2]. This increased substitution rate allows for the identification of a greater proportion, achieving ~22% percent of transcriptome, of zygotic genes with a cutoff of 70% new-to-total ratio, offering increased flexibility in experimental design by eliminating the dependence on fresh prepared samples. Additionally, the expression of *prnc2* in newly identified zygotic genes was experimentally validated, further confirming the reliability of this chemical conversion method.

Our previous development of scNT-seq integrated TFEA-based method with the home-brew Drop-seq platform, enabling on-beads conversion of captured RNA attached to gel beads before reverse transcription. However, this method is incompatible with the 10× Genomics platform[32], because reverse transcription is performed during cell/bead co-encapsulation step. To address this, a recent study on zygotic genome activation in zebrafish employed an optimized IAA-based on-beads conversion step using the home-brew Drop-seq platform[7]. While effective, the lower capture rate of the Drop-seq platform (~5%) poses challenges when working with limited input material, such as during early embryo development, especially in the critical first wave of zygotic genome activation[45,46]. To overcome this limitation, in-situ conversion using the commercial 10× Genomics platform has been implemented due to its higher capture efficiency[8]. However, our direct comparison showed that on-beads conversion is more efficient than in-situ conversion when using the same method. Notably, our analysis highlighted that the inefficient in-situ reaction may lead to an underestimation of zygotic genes (Fig. 3d, e and Supplementary Fig. 7b). To achieve both high capture efficiency and effective T-to-C substitution, we benchmarked widely used commercial platforms and found that the MGI C4 platform, when paired with on-beads chemical conversion, offers the highest sensitivity. This approach requires fewer input cells while providing superior capture efficiency, higher T-to-C substitution rates, and enhanced detection of

genes and UMIs per cell (Fig. 4). Although the barcoded beads used in the MGI C4 platform are incompatible with TFEA-based method, they are fully compatible with IAA-based method without compromising sensitivity. Our findings provide valuable guidance for selecting the appropriate scRNA-seq platform and chemical conversion method for large-scale single-cell RNA labeling. Additionally, our optimized chemical conversion methods are compatible with emerging single-cell platforms like PIP-seq[47], which supports on-beads chemical conversion, further extending the applicability of metabolic RNA labeling in high-throughput single-cell transcriptomics. Furthermore, Well-TEMP-seq[5], which also utilizes barcoded beads similar to Drop-seq and scNT-seq, could benefit from our optimized on-beads chemical conversion to enhance sensitivity.

ZGA is temporally and spatially coordinated process essential for embryonic development[8,48]. While its timing in zebrafish appears independent of cell cycle length, studies[40,42] and our data suggest that cell cycle lengthening enhances transcriptional competence, promoting lineage-specific gene activation. By integrating single-cell metabolic labeling with a Chk1 overexpression-induced cell cycle arrest model, we demonstrated that this approach provides a refined perspective on how cell cycle dynamics shape cell-type-specific gene expression during early embryogenesis. These insights highlight the value of our optimized metabolic RNA labeling-based scRNA-seq for studying ZGA and cell cycle regulation in vertebrate development. Furthermore, our optimized chemistry, readily adaptable to spatial transcriptomics, enables precise mapping of newly synthesized transcripts across embryonic tissues. With enhanced sensitivity, it offers a powerful tool for advancing spatial ZGA profiling and uncovering new insights into lineage-specific gene activation and embryonic patterning.

While our optimized chemical conversion methods enhance the sensitivity and transcript recovery of 4sU-based metabolic labeling, our approach is currently limited to poly(A)-tailed mRNAs, excluding key non-coding RNAs (e.g., microRNAs, lincRNAs, and endogenous retroviral elements) that play critical roles in early embryogenesis[40,49,50]. Integrating poly(A)-independent single-cell RNA-seq methods, such as Smart-seq-total[51] or snRandom-seq[52], could provide a more comprehensive transcriptomic view. Additionally, direct RNA sequencing approaches[53] bypass chemical conversion entirely, offering an alternative for tracking RNA modifications, which could complement 4sU-based techniques.

Beyond standard single-cell RNA-seq, our optimized chemical conversion conditions can be integrated with other platforms and technologies to expand its utility. scGRO-seq[54], a single-cell adaptation of GRO-seq, employs click chemistry to capture actively transcribing RNA polymerases, providing high-resolution insights into transcriptional burst kinetics and nascent RNA synthesis. Combining 4sU labeling with scGRO-seq could offer a more comprehensive view of transcriptional dynamics by distinguishing between newly

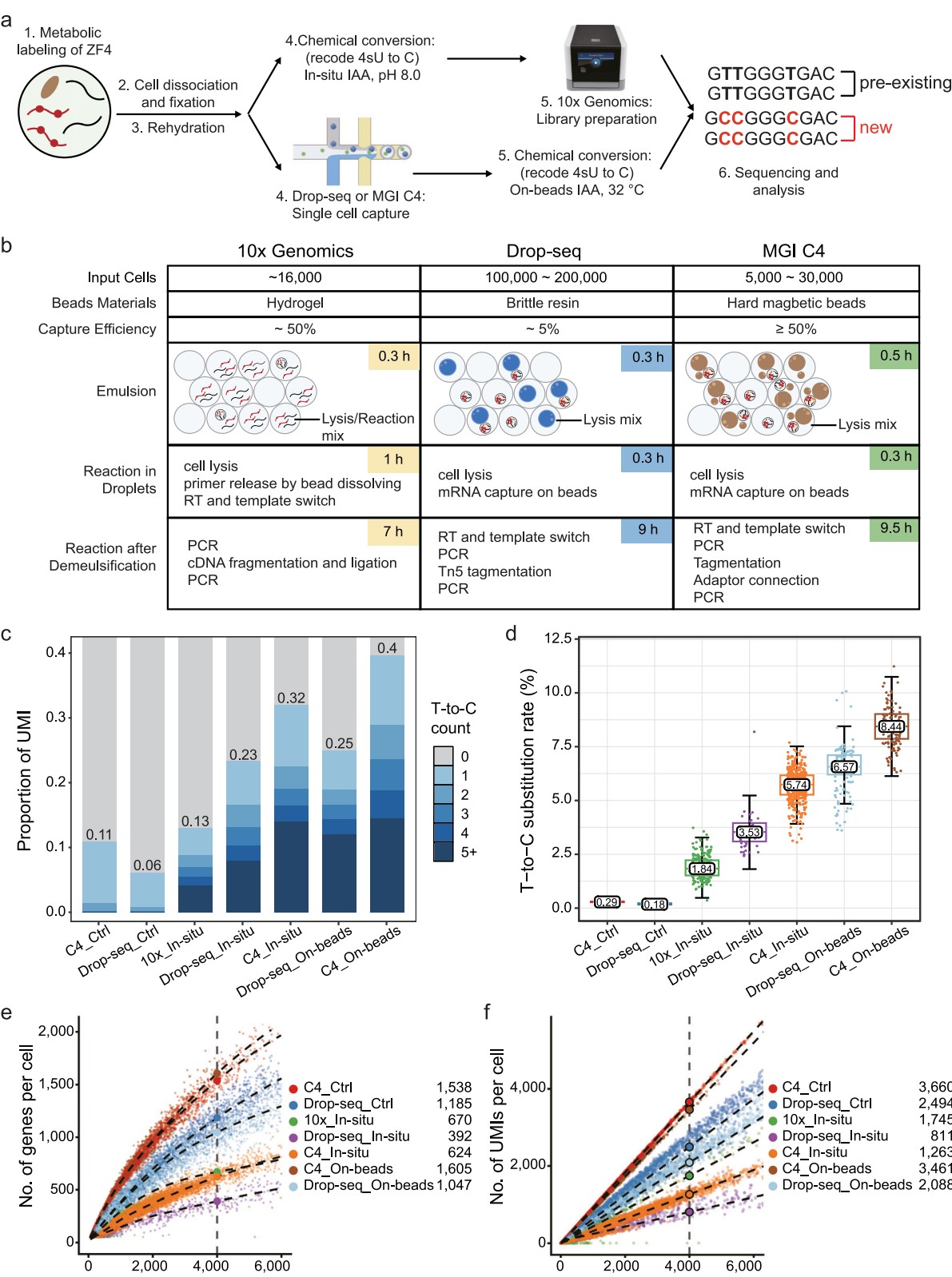

synthesized transcripts and ongoing transcription activity. While the compatibility of 4sU labeling with click chemistry requires further validation, a more practical approach would be to apply metabolic labeling scRNA-seq methods, such as scNT-seq, alongside scGRO-seq on the same samples and integrate the data computationally to capture both newly synthesized transcripts and actively transcribing RNA polymerases.

In conclusion, our comparison of ten key methods for metabolic labeling in scRNA-seq demonstrated that the mCPBA/TFEA-based method is optimal for quantifying transcriptomes from large cell populations (e.g., >100,000 cells), especially considering potential cell loss during handling steps like centrifugation and washing. Conversely, the on-beads IAA-based method, compatible with scRNA-seq platforms utilizing barcoded beads for mRNA capture, allows efficient

**Fig. 4 | Comparison between 10× Genomics, Drop-seq and MGI C4 high-throughput single-cell platforms. a** Overview of metabolic labeling high-throughput scRNA-seq using 10× Genomics, Drop-seq, and MGI C4 platforms in ZF4 cells. Both platforms are capable of performing on-beads chemical conversion reactions during the library preparation steps. **b** Schematic comparison of the three high-throughput scRNA-seq platforms, highlighting differences in input cell quantity, beads materials, capture efficiency, time consumption, and key steps involved in library preparation. RT reverse transcription. **c** Proportion of UMIs containing T-to-C substitutions under different conditions and platforms. The color gradient indicates the number of T-to-C substitutions per read, with darker shades representing a higher number of substitutions within the UMI. "Ctrl" represents the control group without chemical treatment; "In-situ" refers to "In-situ IAA, pH8.0" method, while "on-beads" indicates "On-beads IAA, 32 °C" chemistry in (**c**–**f**).

Source data are provided as a Source Data file. **d** Box plot showing T-to-C substitution rates across the scRNA-seq platforms. Different colored boxes represent various platforms and treatment methods. The box edges correspond to the 25th and 75th percentiles, with the *x*-axis displaying types of base substitutions. Ctrl represents the control sample without chemical treatment. Source data are provided as a Source Data file. **e**, **f** Scatterplots showing the number of genes (**e**) or UMIs (**f**) detected per cell as a function of aligned reads per cell across the different platforms. Different colored dots represent various platforms and treatment methods. Fitted lines and predicted numbers of genes or UMIs detected per cell at 4000 reads are shown for each platform. The predicted values for 4000 reads are displayed in the upper left corner of the figure. The curve in (**e**) is smoothed using locally weighted regression, while in (**f**) is smoothed using a linear model.

buffer exchange for chemical conversion on beads, as seen in MGI C4[35] and Well-Paired-seq[34]. Leveraging this commercial platform enhances transcriptome analysis, particularly for low-input cell samples (e.g., <10,000 cells). This analysis provides a robust framework for making informed decisions about the most appropriate methods and sets the stage for future advancements in time-resolved scRNA-seq methodologies (Supplementary Fig. 11). These insights will deepen our understanding of RNA dynamics and enhance the interpretation of RNA regulation in diverse biological systems.

## Methods

### Ethics
All fish were maintained and experiments conducted in accordance with protocols reviewed and approved by the Animal Ethics Committee on Laboratory Animal Care and Use of Shanghai Ocean University, with approval granted on February 26, 2022.

### ZF4 cell cultures and 4sU metabolic labeling
ZF4 cell lines were cultured in DMEM/F12 (Gibco, C11330500BT) medium supplemented with 10% fetal bovine serum (FBS) (Gibco, A5669701) and 1% penicillin-streptomycin (Gibco, 15140-122). The cells were maintained at 28 °C with 5% $CO_2$ and passaged every 3 to 4 days. For labeling experiments, 4-thiouridine (4sU) (Alfa Aesar, J60679) was dissolved in dimethylsulfoxide (DMSO) (Sigma-Aldrich, D5879-500ML) to create a 1 M stock solution. Two days before labeling, ZF4 cells were seeded at a density of $2 \times 10^5$ cells per mL. The 4sU labeling was performed by incubating ZF4 cells in fresh medium supplemented with 4sU at a final concentration of 100 μM, following previously established protocols[55]. After 4 h of labeling, the cells were rinsed once with DPBS (BBI, C14190500BT) and dissociated into single-cell suspensions using 0.25% Trypsin-EDTA (Gibco, 25200056) for 2 to 3 min at 28 °C.

### Zebrafish maintenance
Fish are maintained at 28 ± 0.2 °C on a 14-h light/10-h dark cycle. Animals up to 5 days post-fertilization (dpf) are maintained in culture medium (5 mM NaCl (BBI, A501218-0001), 0.17 mM KCl (BBI, A501159-0500), 0.33 mM $CaCl_2$ (BBI, A100556), 0.33 mM $MgSO_4$ (BBI, A601988-0250), and 0.1% Methylene blue (BBI, A610622-0025)). Embryos were grown and staged according to standard procedures[56]. Zebrafish embryos from wild-type AB and TU strains were used for all experiments. Sex was not considered, as it is difficult to determine and not relevant at the early embryonic stages analyzed in this study.

### Embryo microinjection and single-cell suspension preparation
Fertilized eggs were collected and maintained in culture medium at 28 °C. One-cell stage wild-type zebrafish embryos were microinjected with 1 nL of 50 mM 4sU containing 1 ng/μL non-toxic fluorescent dextran dye (Invitrogen, D1821) to visually confirm the successful injection of 4sU under fluorescence microscopy. At 5.5 h post-fertilization (hpf), 200 injected embryos were randomly collected per sample after

visually confirming they had reached the expected developmental stage. Embryos were dechorionated by treatment with 1 mg/mL protease from *Streptomyces griseus* (Sigma-Aldrich, P8811-1G) in 60 mm dishes coated with 2% agarose (Biowest, BY-R0100; dissolved in water). The dechorionated embryos were then placed in 1 mL of ice-cold deyolking buffer (55 mM NaCl, 1.8 mM KCl, and 1.25 mM $NaHCO_3$ (BBI, A610482-0500)) in an Eppendorf tube and gently pipetted up and down about 10 times with a P1000 tip until the yolk was dissolved. The tube was then centrifuged at $500 \times g$ for 1 min at 4 °C. The final pellet was resuspended in 200 μL DPBS containing 0.01% bovine serum albumin (BSA) (Sigma-Aldrich, A8806-5G).

### Flow cytometry analysis of cell cycle states in ZF4 cells
After 4sU labeling, ZF4 cells were dissociated into single cell suspension as described above. The cells were counted, and aliquots of 1 million cells were dispensed into labeled tubes, followed by washing with DPBS and centrifugation for 5 min at $100 \times g$. For fixation, the cell pellet was resuspended in 300 μL PBS with gentle vortex, followed by the addition of 700 μL ice cold ethanol dropwise while continuously vortexing. The mixture was then incubated at 4 °C for 30 min to overnight. After fixation, the cells were centrifuged and resuspended in 250 μL of DPBS followed by the addition 5 μL of 10 mg/mL RNase A (Sigma-Aldrich, R-6513) to a final concentration of 0.2–0.5 mg/mL. The cells were then incubated at 37 °C for one hour. Cells were then stained by adding 10 μL of a 1 mg/mL propidium iodide (PI) solution (Sigma-Aldrich, P-4170) to a final concentration of 10 μg/mL, and kept in the dark for at least one hour until analysis. The cell cycle was analysed using the BD FACSMelody (BD Biosciences). Fractions of cells in each phase were quantified using FlowJo software V10.8.1 (Supplementary Fig. 3d). During quantification, lymphocytes were identified by gating on forward-scatter area (FSC-A) and side-scatter area (SSC-A). Doublets were excluded using FSC-A versus forward-scatter height (FSC-H). Cohesive cells were further exclude using PE-A versus PE-H (adjust the PE coordinates to linear). Using the Cell Cycle plug-in to conduct cell cycle analysis: select the appropriate channel (PE-A) for the horizontal coordinate and select the appropriate fitting algorithm Watson (Pragmatic) in the model.

### Cell fixation, cryopreservation, and rehydration for sample processing
After 4sU labeling, ZF4 cells or zebrafish embryo cells were dissociated into single cell suspension as described above. For cell fixation and cryopreservation, the dissociated cells were then resuspended in 200 μL of DPBS containing 0.01% BSA, and 800 μL of cold methanol (Sigma-Aldrich, 34860-4L-R) was added dropwise, resulting in a final concentration of 80% methanol in DPBS. The cell suspension was mixed and incubated on ice for one hour, after which the methanol-fixed cells were stored at −80 °C for up to one month.

For sample rehydration, cells were taken out from −80 °C and kept on ice throughout the entire procedure. The cells were centrifuged at $1000 \times g$ for 5 min at 4 °C, and the methanol-DPBS solution

was carefully removed. The cells were then resuspended in 1 mL of rehydration buffer (0.01% BSA in DPBS supplemented with 1 U/μL RNase inhibitor (Lucigen, F83923-1) and 100 mM Dithiothreitol (DTT, Thermo Scientific, R0862)). Following another centrifugation at 1000 × g for 5 min at 4 °C, the rehydration buffer was removed, and the cells were resuspended in 1 mL of DPBS containing 0.01% BSA. The cell suspension was passed through a cell sieve and counted. The single-cell suspension was then diluted to a concentration of either 100 cells per μL for Drop-seq or 800–2000 cells per μL for MGI C4, and immediately used for downstream processing.

## Single cell RNA library preparation and sequencing

For 10× Genomics, the cells underwent chemical conversion using in-situ IAA-based methods to convert 4sU into a cytidine analog prior to single cell capture by 10× Genomics (see details in the "Chemical conversion reactions" section). After chemical conversion, we performed all steps following the 10× protocol. Cellular suspensions and other reagents were loaded into the chip (10× Genomics). Then we ran the Chromium Controller for single-cell GEMs generation. Single-cell RNA-Seq libraries were prepared using Chromium Next GEM Single Cell 3' Reagent Kits v3.1 (10× Genomics). GEM-reverse transcription (RT) was performed in a C1000 Touch Thermal cycler with 96-Deep Well Reaction Module: 53 °C for 45 min, 85 °C for 5 min; held at 4 °C. After RT, GEMs were broken, and the single-strand cDNA was cleaned up with DynaBeads MyOne SILANE. Then cDNA was amplified using the C1000 Touch Thermal cycler with 96-Deep Well Reaction Module: 98 °C for 3 min; cycled 11×: 98 °C for 15 s, 63 °C for 20 s, and 72 °C for 1 min; 72 °C for 1 min; held at 4 °C. Amplified cDNA product was cleaned up with 0.6× SPRIselect Reagent (Beckman Coulter, B23318). Then cDNA libraries were constructed using the reagents in the Chromium Next GEM Single Cell 3' Library Kit v3.1, following these steps: (1) fragmentation, end repair and A-tailing; (2) post fragmentation end repair and A-tailing double sided size selection with SPRIselect; (3) adapter ligation; (4) post ligation cleanup with SPRIselect; (5) sample index PCR and cleanup. After quality control using a Bioanalyzer 2100 (Agilent, 5067–4626), libraries were sequenced on an Illumina HiSeq X Ten instrument (Illumina).

For Drop-seq, cell and barcoded bead co-encapsulation, library preparation, and sequencing were performed[55]. Specifically, the single-cell suspension was counted and diluted to a concentration of 100 cells per μL in PBS containing 0.01% BSA. The flow rates for cells and beads (Chemgenes, MACOSKO-2011-10) were set to 4000 μL per hour, while QX200 droplet generation oil (Bio-rad, 1864006) was run at 15,000 μL per hour. After droplet breakage, the beads underwent chemical conversion using SLAM-seq or TimeLapse-seq methods to convert 4sU into a cytidine analog (see details in the "Chemical conversion reactions" section). Reverse transcription, library preparation, and sequencing were then performed. In brief, for up to 120,000 beads, 200 μL of reverse transcription mix (1× Maxima reverse transcription buffer, 4% Ficoll PM-400 (Sigma, F4375-25G), 1 mM dNTPs (Clontech, 639125), 1 U/μL RNase inhibitor (Lucigen, 30281-2), 2.5 μM template switch oligo (TSO: AAGCAGTGGTATCAACGCAGAGTGAATrGrGrG), and 10 U/μL Maxima H Minus reverse transcriptase (Thermo Fisher, EP0753)) were added. The reverse transcription reaction was carried out at room temperature for 30 min and then at 42 °C for 2 h. After Exonuclease I treatment (NEB, M0293L), pooled beads were washed once with TE-SDS buffer (10 mM Tris-HCl pH 8.0 (Invitrogen, 15568-025), 1 mM EDTA, and 0.5% SDS) and twice with TE-TW buffer (10 mM Tris-HCl pH 8.0, 1 mM EDTA, and 0.01% Tween-20). The beads were resuspended in 500 μL of 0.1 M NaOH and incubated for 5 min at room temperature with rotation. To neutralize, 500 μL of 0.2 M Tris-HCl pH 7.5 (Invitrogen, 15567-027) was added. The beads were washed once with TE-TW buffer and once with 10 mM Tris-HCl pH 8.0. Second-strand synthesis was performed to increase library recovery. For the second-strand synthesis reaction, the beads were resuspended in

200 μL of reaction mixture (1× Isothermal Amplification Buffer, 4% Ficoll PM-400, 1.4 mM dNTPs, 6 mM MgSO₄, 10 μM TSO-N3G2N4B (AAGCAGTGGTATCAACGCAGAGTGANNNGGNNNNB; $N$ = A, T, C, G at a 25:25:25:25 ratio), and 0.4 U/μL Bst 3.0 DNA polymerase (NEB, M0374)). The reaction was incubated at 60 °C for 15 min with rotation and then stopped by washing the beads once with TE-SDS buffer and twice with TE-TW buffer.

To determine the optimal number of PCR cycles for cDNA amplification, PCR reactions (~6000 beads per tube) were prepared in 50 μL volumes (25 μL of 2× KAPA HiFi HotStart ReadyMix (KAPA Biosystems, KK2602), 0.4 μL of 100 μM TSO-PCR primer (AAGCAGTGGTATCAACGCAGAGT), and 24.6 μL of nuclease-free water). Full-length cDNA was amplified with the following thermal cycling conditions: 95 °C for 3 min; 4 cycles of (98 °C for 20 s, 65 °C for 45 s, 72 °C for 3 min); 9–12 cycles of (98 °C for 20 s, 67 °C for 45 s, 72 °C for 3 min); 72 °C for 5 min; held at 4 °C. cDNA was then tagmented using the Nextera XT DNA Sample Preparation Kit (Illumina, FC-131-1096), starting with 550–600 pg of pooled cDNA from all PCR reactions. The library was further amplified with 12 enrichment cycles using Illumina Nextera XT i7 primers and the P5-TSO hybrid primer (AATGATACGGCGACCACCGAGATCTACACGCCTGTCCGCGGAAGCAGTGGTATCAACGCAGAGTAC). After quality control using a Bioanalyzer 2100 (Agilent, 5067–4626), libraries were sequenced on an Illumina HiSeq X Ten instrument (Illumina). Custom Read 1 Primer (GCCTGTCCGCGGAAGCAGTGGTATCAACGCAGAGTAC) was added.

For MGI C4, scRNA-seq libraries were prepared using the DNBelab C Series Single-Cell Library Prep set (MGI, 940-001818-00, https://www.mgi-tech.com//Uploads/Temp/file/20240329/66066e0f968cd.pdf)[35]. Single-cell suspensions were converted to barcoded scRNA-seq libraries through droplet encapsulation and emulsion breakage. After droplet breakage, the beads underwent chemical conversion using the on-beads IAA reaction (see details in the "Chemical conversion reactions" section). The beads were then washed once with 1× SSC, followed by reverse transcription, enzyme digestion, second-strand synthesis, PCR amplification of the cDNA library, and indexed sequencing libraries according to the manufacturer's protocol. Library concentration was quantified using a Qubit dsDNA Assay Kit (Thermo Fisher Scientific, Q32854). Final libraries were sequenced on MGI DNBSEQ-T7 sequencer. The read structure was paired-end with Read 1 (30 bases) covering the 10-bp cell barcode 1, 10-bp cell barcode 2, and 10-bp unique molecular identifier (UMI), and Read 2 (100 bases) containing the transcript sequence.

## Chemical conversion reactions

For in-situ IAA, chemical conversion was conducted after cell fixation and prior to cell rehydration (see details in the "Cell fixation, cryopreservation, and rehydration for sample processing" reaction):

1) for the in-situ IAA pH 7.5 method[8,9], cells were first removed from −80 °C on to ice, and 1 M iodoacetamide (IAA) (Sigma-Aldrich, I1149-25G) solution was added in 80% methanol and 20% DPBS containing 0.01% BSA to a final concentration of 10 mM, incubating the mixture on ice for 15 min, followed by incubation at 50 °C for 15 min;

2) for the in-situ IAA pH 8.0 method[3,4], cells were first removed from −80 °C on to ice for 3 mins and gently resuspended. Cells were then spun down at 2000 × g, 4 °C for 5 min and resuspended in 1 mL DPBS containing 0.1% DEPC, 3% BSA, and 10 mM DTT, spun again and resuspended in 100 μL DPBS containing 3% BSA. To this suspension, 220 μL water, then 40 μL NaPO₄ buffer, then 40 μL 100 mM IAA were added. Cells were incubated at 50 °C for 15 min, being gently resuspended every five minutes;

For on-beads IAA, chemical conversion was conducted after droplet breakage and prior to the reverse transcription (see details in the "Single cell RNA library preparation and sequencing" section):

3) for on-beads IAA 32 °C[7,9], the beads were washed once with 300 μL of 5× IAA buffer (250 mM NaPO$_4$, pH 8.0) and then incubated in 300 μL of IAA reaction mixture (10 mM IAA, 50 mM NaPO$_4$, pH 8.0, 20% DMSO and 6% Ficoll PM-400 (Sigma, F4375-25G)) for 15 min at 32 °C. The reaction was stopped by adding 6 μL of 1 M DTT to a final concentration of 20 mM;

4) for on-beads IAA 37 °C[5,6], the beads were resuspended in reaction buffer (50 mM DPBS, 10 mM IAA, 10% DMSO, pH 8.0) and incubated at 37 °C for 1 h. Then, 10 mM DTT was added to stop the reaction;

For on-beads TimeLapse[10,36,55], following droplet breakage, beads were washed with 300 μL of 6× SSC and then incubated in 250 μL of the respective reaction mixture for 1 h at 45 °C:

5) for the mCPBA/TFEA pH 5.2 method, 600 mM 2,2,2-tri-fluoroethylamine (TFEA) (MREDA, M014297), 1 mM EDTA (Invitrogen, 15575-038), and 100 mM sodium acetate pH 5.2 (Invitrogen, AM9740), with 10 mM meta-chloroperoxy-benzoic acid (mCPBA) (Alfa Aesar, AAAL00286-14);

6) for the mCPBA/TFEA pH 7.4 method, 600 mM TFEA, 1 mM EDTA, and 100 mM Tris-HCl pH 7.4, with 10 mM mCPBA;

7) for the NaIO$_4$/TFEA pH 5.2 method, 600 mM TFEA, 1 mM EDTA, and 100 mM sodium acetate pH 5.2, with 10 mM sodium periodate (NaIO$_4$) (Across Organics, AC419610050);

8) for the NaIO$_4$/TFEA pH 7.4 method, 600 mM TFEA, 1 mM EDTA, and 100 mM Tris-HCl pH 7.4, with 10 mM NaIO$_4$;

9) for the NaIO$_4$/NH$_4$Cl pH 8.8 method, 600 mM ammonium chloride (NH$_4$Cl) (Sigma, 09718), 1 mM EDTA, and 100 mM Tris-HCl pH 8.8, with 10 mM NaIO$_4$;

After the on-beads TimeLapse-based reaction, beads were washed with 500 μL of TE buffer (10 mM Tris-HCl pH 8.0, 1 mM EDTA pH 7.5) and incubated in 500 μL of reducing buffer (10 mM DTT, 10 mM Tris-HCl pH 7.5, 1 mM EDTA, 100 mM NaCl, and 0.8 U/μL RNase Inhibitor) for 30 min at 37 °C.

For on-beads TUC-seq[30,36], following droplet breakage, beads were washed with 300 μL of 6× SSC and then incubated in 250 μL of the reaction mixture for 3 h at 50 °C:

10) for the OsO$_4$/NH4Cl pH 8.8 method, 180 mM NH4Cl and 0.45 mM OsO$_4$.

## Whole-mount in situ hybridization of zebrafish embryos

Whole-mount in situ hybridization (WISH) in zebrafish embryos (Fig. 3g and Supplementary Fig. 8c) was performed according to the published protocol[57]. Briefly, DNA templates were amplified from cDNA obtained from wild-type embryos at 5.5 hpf using the primers synthesised in Sangon Biotech (Shanghai, China) (see Supplementary Table 4). Antisense and sense-strand (as negative control) RNA probes were labeled with digoxigenin-linked nucleotides (Roche, 11277073910) and transcribed in vitro using T7 RNA polymerase (Thermo Fisher, AM1320). The probes were subsequently purified by LiCl (Invitrogen, 9480G)/ethanol (Sangon Biotech, A500737-0500) precipitation. Zebrafish embryos intended for WISH were collected at 5.5 hpf and fixed overnight in 4% paraformaldehyde (BBI, E672002-0500). After fixation, embryos were washed with PBS containing 0.1% Tween-20 (Sigma-Aldrich, P7949-500ML), dehydrated in 100% methanol for 15 min at room temperature, and stored at −20 °C for at least 24 h. After rehydration, embryos were incubated in hybridization solution (50% Formamide (Thermo Fisher, 17899), 5× SSC (Corning, 46-020-CM), 50 μg/mL heparin (BBI, A603251-0001), 500 μg/mL tRNA (Sigma, R6750-100G), 0.1% Tween-20) at 70 °C. Following 4 h of pre-hybridization, 100 ng of RNA probe was added, and hybridization was conducted overnight at 70 °C. Embryos were then washed and blocked in a solution of 2 mg/mL BSA, 2% sheep serum, and 0.1% Tween-20 for 3 h, then incubated overnight with anti-digoxigenin (DIG) antibody (1:5000 in blocking solution) (Roche, 11093274910). Finally, embryos were washed and stained with a solution of 0.321 mg/mL Nitro Blue

Tetrazolium (Sigma, N6639) and 0.175 mg/mL 5-bromo-4-chloro-3-indolyl phosphate (Sigma, B-8503) in alkaline Tris buffer (100 mM Tris-HCl, pH 9.5, 50 mM MgCl$_2$, 100 mM NaCl, 0.1% Tween-20). After staining, embryos were washed and imaged using a ZEISS Axio Zoom V16 fluorescence microscope.

## Read alignment and quantification of metabolically labeled transcripts

DropSeq datasets. For the raw paired-end sequencing data from Drop-seq experiments, each mRNA read was uniquely identified by combing a cell barcode (bases 1–12 of Read 1) and a unique molecular identifier (UMI, bases 13–20 of Read 2). Sequencing adapters and polyA tails form Read 2 were trimmed using fastp v0.20.1 with the parameters "−trim_poly_x, --adapter_sequence TCTTTCCCTACACGACGCTCTT CCGATCT"[58], while sequences beyond the initial 20 bases of Read 1 were removed using cutadapt v4.2[59] (parameters: -u −105). The remaining sequences from Read 1 were aligned to the zebrafish genome assembly (GRCz11, Ensemble release 108) using dynast v1.0.1[37](https://github.com/aristoteleo/dynast-release/) align command (parameters: −soloCellFilter CellRanger2.2 "predicted number of cells in the experiment" 0.99 10). Reads mapped to exonic or intronic regions of annotated genes on the predicted strand were retained for downstream analyses.

To quantify labeled and unlabeled RNA, we used the dynast v1.0.1 count command (parameters: −conversion TC, −barcode-tag CB, −umi-tag UB), generating the rates.csv file for calculating T-to-C substitution as well as other base substitution rates (Figs. 2a, 4d and Supplementary Fig. 2a, 7d). T-to-C substitutions with a Phred quality score > 27 were retained for quantification. The proportion of labeled RNA was calculated using the labeled_TC and unlabeled_TC layers from the adata.h5ad file in dynast, enabling the identification of labeled and unlabeled RNA (Supplementary Fig. 2b).

MGI C4 datasets. For the raw paired-end sequencing data from the MGI C4 libraries, DNBC4tools (https://pypi.org/project/DNBC4tools/) and dynast v1.0.1 were used. Each mRNA read was uniquely identified by a combination of two cell barcodes (located within bases 6–15 and 22–31 of Read 1) and a UMI (bases 37–46 of Read 2). First, DNBC4tools was employed to generate a barcode whitelist, and the barcode/UMI positions were defined in the parameter settings of dynast software. Subsequently, the align and count command were run (align parameters: −soloCBmatchWLtype 1MM −soloCBwhitelist whitelist1 whitelist2, count parameters: −conversion TC, −barcode-tag CB, −umi-tag UB), to calculate T-to-C substitutions (Fig. 4d and Supplementary Fig. 7d), and generated labeled and unlabeled count matrices based on the adata.h5ad file from the dynast output.

For the correlation analysis, we first retrieved the expression values for both labeled and unlabeled RNA. These values were then log-transformed, and pairwise Pearson correlation coefficients were calculated using cor function in R (Supplementary Fig. 4a), scatter plots were generated with the pairs function in R (parameters: gap = 0, lower.panel = custom function, pch = 20). The custom function used for this visualization is detailed in the "Data and code availability" section.

To calculate the proportion of UMIs containing T-to-C substitutions (Fig. 4c and Supplementary Fig. 2c), we quantified the T-to-C substitution UMI count for each cell from counts_TC.csv file and calculated the proportion of labeled UMIs per cell, visualizing the data using ggplot2 v3.4.4 (https://github.com/tidyverse/ggplot2). To visualize T-to-C substitutions in transcripts with unique UMIs for a specific gene in a cell, we used the scNT-seq Perl script (https://github.com/wulabupenn/scNT-seq) to extract the gene, UMI, and read information across all cells. For single-gene visualization, a custom R function was applied to process the sequencing data, calculate base changes for each UMI, and mark T-to-C substitution events. Scatter plots were then generated to display these substitutions across

genomic positions for different UMIs (Fig. 2e and Supplementary Fig. 5a, b).

## Curve fitting and prediction of detected genes and transcripts based on sequencing depth

To predict the number of detected genes (nGene) and transcripts (nUMI) across samples, we applied both locally weighted regression (LOESS) and linear regression (LM) models. For nGene predictions, we used the loess function from the R stats package with the following parameters: formula = nGene ~ reads, span = 0.75, degree = 2, method = "loess". We selected cells with more than 100 detected genes from ZF4 cell line samples and more than 200 detected genes from 5.5 hpf zebrafish embryo samples. Based on the fitted LOESS model, we predicted the number of genes detected at 10,000 reads for ZF4 samples (Fig. 2b) and at 4000 reads for zebrafish samples (Fig. 4e and Supplementary Fig. 10a).

For nUMI predictions, we employed the lm function with the parameters: formula = nUMI ~ reads, method = "qr", model = TRUE. Cells with more than 400 UMIs were selected from both ZF4 and zebrafish samples. Linear regression was applied to assess the relationship between UMI counts and read counts, predicting nUMI detection at 10,000 reads for ZF4 samples (Fig. 2c) and 4000 reads for zebrafish samples (Fig. 4f and Supplementary Fig. 10b).

## Calling consensus sequences and filtering SNPs

To handle reads from different regions of the same mRNA, consensus sequences were generated using the dynast consensus procedure with default parameters. Samples without chemical conversion served as control to exclude SNPs that could interfere with accurate T-to-C substitution identification, SNPs were detected using the dynast count command (parameters: –control, –snp-threshold 0.5, –conversion TC), and background substitution rates for unlabeled RNA were calculated with the estimate command (parameter: –control), outputting to p_e.csv. Finally, labeled and unlabeled count matrices were generated using the dynast count command (parameters: –snp-csv, –conversion TC), filtering out SNPs.

## Estimation of the fraction of newly synthesized transcripts

To estimate newly synthesized transcripts, we modeled the distribution of unlabeled and labeled RNA counts using a binomial mixture model[10,12]. We employed dynast for the statistically estimation of new and old transcripts based on labeled and unlabeled UMI counts (parameters: –method alpha, –p-e p_e.csv). The alpha method, derived from the detection rate estimation used in scNT-seq[2], is particularly well-suited for handling sparse data. This approach estimates the substitution rate for both unlabeled and labeled RNA, allowing us to model the fraction of newly synthesized RNA within the total RNA pool. As a result, we were able to categorize mRNA counts into five distinct types: unspliced, spliced, estimated-labeled, estimated-unlabeled, and total RNA matrices.

## Cell-type clustering and dataset integration

For benchmarking the ten chemical conversion methods (Fig. 1a, b) using ZF4 cell line samples, we combined the raw digital expression matrices of labeled and unlabeled UMI counts. These matrices were processed using the R package Seurat v4.3.0[60]. Quality control measures were applied to filter out low-quality cells, including those with mitochondrial UMI counts ≥10% or those with fewer than 400 or more than 4000 detected genes. After these filters were applied, 41,240 cells remained for downstream analysis.

To integrate the single-cell RNA sequencing data from different chemical treatments and untreated controls (Fig. 2d and Supplementary Fig. 3b), we employed the Canonical Correlation Analysis (CCA) method within the Seurat package, using the FindIntegrationAnchors and IntegrateData functions. Each sample was normalized independently, and highly variable genes were identified using the FindVariableFeatures function (parameters: selection.method = vst and nfeatures = 2000). CCA was then performed on these 2000 highly variable genes to reduce dimensionality, selecting the top 10 principal components (PCs) for further analysis. UMAP was used for visualization via the RunUMAP function (parameters: umap.method = umap-learn). Clustering was performed using the FindCluster function (parameters: resolution = 0.02), and marker genes for each cluster were identified using the FindAllMarkers function (parameters: min.pct = 0.25, logfc.threshold = 0.25, test.use = "wilcox"). Seurat makes available a list of cell cycle marker genes, and then we performed homologous gene conversion for these marker genes in Ensembl Biomart, extracted the corresponding gene expression levels from the normalized data using the AverageExpression function, and visualized the data through a heatmap using the ComplexHeatmap v2.14.0 package. To assess whether there were significant differences in the number of dividing cells identified by different methods, a Wilcoxon rank-sum test was performed. The results were visualized using a box plot, providing a comparative overview of the distribution of dividing cell counts across methods (Supplementary Fig. 3c).

For the 5.5 hpf zebrafish embryo samples, raw digital expression matrices of labeled and unlabeled UMI counts were processed using Python. Preprocessing was executed with omicverse v1.5.7 (https://github.com/Starlitnightly/omicverse)[61]. Quality control was performed using the qc function from the OmicVerse package (parameters: tresh = {'mito_perc': 0.1, 'nUMIs': 800, 'detected_genes': 400}, mt_startswith = 'mt-'). Cells with mitochondrial gene expression ≥10%, UMI counts <800, or fewer than 400 detected genes were excluded. Potential doublets were identified and removed.

Principal component analysis (PCA), neighborhood graph computation, and neighborhood graph embedding were conducted using Scanpy v1.9.6[62]. The dataset was normalized using scanpy.pp.normalize_total (parameters: target_sum = 1e4), followed by logarithmic transformation with scanpy.pp.log1p. Highly variable genes were identified using scanpy.pp.highly_variable_genes (parameters: min_mean = 0.0125, max_mean = 3, min_disp = 0.5). Dimensionality reduction was performed with scanpy.tl.pca (parameter: svd_solver = 'arpack'), and principal components were ranked based on variance using sc.pl.pca_variance_ratio (parameter: log = True), selecting the top 16 PCs. The neighborhood graph was computed using sc.pp.neighbors (parameters: n_neighbors = 10, n_pcs = 16, use_rep = 'X_pca').

To correct for batch effects from different experimental conditions, we applied Harmony integration using scanpy.external.pp.harmony_integrate with default parameters[63]. UMAP clustering and visualization were performed using scanpy.tl.umap (parameters: n_components = 2, method = 'umap') (Fig. 3b). Cluster-specific marker genes were identified with the rank_genes_groups function in Scanpy (parameters: Wilcoxon rank-sum test) (Fig. 3c). Genes with an adjusted P-value < 0.05 (Bonferroni corrected) were considered differentially expressed.

## Gene Ontology and KEGG pathway enrichment analysis

Functional enrichment analysis (Supplementary Fig. 4b) of differentially expressed genes was performed using R package clusterProfiler v4.6.2, based on hypergeometric distribution[64]. GO term enrichment was performed using the enrichGO function (parameters: OrgDb = org.Dr.eg.db, ont = 'ALL', pAdjustMethod = 'fdr'). For KEGG pathway enrichment, zebrafish gene symbols were converted to entrezids using the toTable function (parameters: org.Dr.egSYMBOL), followed by pathway enrichment using the enrichKEGG function (parameters: gene = ENTREZID, organism = 'dre', keyType = 'kegg', pAdjustMethod = 'fdr'). GO terms or KEGG pathways with FDR < 0.05 were considered significantly enriched (Supplementary Fig. 6).

### New-to-total RNA ratio (NTR) partition for identifying zygotically activated transcripts

We calculated the new-to-total RNA ratio (NTR) by dividing the number of labeled UMIs detected for each gene by the total number of UMIs for the same gene in each cell. To benchmark the distribution of maternal and zygotic genes across three chemical conversion methods, on-beads IAA, mCPBA/TFEA (pH 5.2 and pH 7.4), we retained 1711, 2953, and 3654 cells, respectively, from the methods after passing filtering criteria set by Omicverse. Genes with fewer than 10 UMIs were filtered out. For each dataset, the retained genes were divided into ten equally sized bins. The data were visualized using violin plots (Supplementary Fig. 7a). Following the gene classification criteria based on the proportion of new RNAs from a published study[7], genes in the top bins were classified as zygotic based on defined NTR thresholds (NTR > 70%, 75%, 80%, and 85%), while those in the bottom bins were classified as maternal (NTR < 5%). For the two published datasets, raw fastq files were downloaded from NCBI under accession numbers GSE224918[7] and GSE158849[8], and processed using the same pipeline as in this study. This allowed for a direct comparison of maternal (M), maternal-zygotic (MZ), and zygotic (Z) genes identified by the three chemical conversion methods (Fig. 3d, e and Supplementary Fig. 7b). Overlap between gene sets with different NTR thresholds was visualized using Venn diagrams generated by ggVennDiagra v1.5.2 (https://github.com/gaospecial/ggVennDiagram), highlighting shared zygotic genes across methods (Fig. 3f and Supplementary Fig. 7c).

### Estimation of RNA half-life

We estimated RNA half-life following the methodology from a previous study based on one time point 4sU labeling experiment[2]. For each gene in the data sets from ZF4 cell line using mCPBA/TFEA (pH 5.2 and 7.4) methods, we separately aggregated the labeled and unlabeled UMI counts in steady-state cells. The fraction of labeled transcripts was calculated by dividing summed labeled UMI counts by the total UMI counts. We defined $n$ as the metabolically labeled RNA, $r$ as the total RNA abundance, $t$ as the labeling time, and $h$ as the RNA half-life for each gene in each cell. The half-life ($h$) can be calculated as:

$$h = \frac{\ln(2)t}{-\ln\left(1 - \frac{n}{r}\right)}$$

Following this, we ranked the half-lives of 17,653 genes from shortest to the longest. The top 10% (unstable) and bottom 10% (stable) of genes were selected for functional enrichment analysis (Supplementary Fig. 6).

### RNA velocity analysis

To compute RNA velocity based on metabolic labeling, we used dynamo v1.4.0 (https://github.com/aristoteleo/dynamo-release)[37]. The preprocessing steps included gene filtering, normalization, highly variable genes identification, principal component analysis (PCA), and neighbor graph construction. RNA velocities were then estimated using the dynamo.tl.dynamics function with "stochastic" dynamical model (parameters: experiment_type = "one-shot", n_top_genes = 3000, NTR_vel = True, re_smooth=True, and model = "stochastic"). High-dimensional velocity vectors were projected into two-dimensional UMAP space using dynamo.tl.reduceDimension function with default parameters and visualized using dynamo's streamline plot. Phase diagrams and randomized streamline plots were generated using the default settings of dynamo.pl.streamline_plot function (Supplementary Fig. 9b). For splicing velocity analysis, we used spliced and unspliced RNA matrices generated by dynast pipeline. RNA velocities were estimated using a "deterministic" dynamical model with parameters set to calc_rnd_vel = True. The downstream processing and visualization steps followed the same procedure as in the metabolic labeling velocity analysis.

### Data from Lee et al. processing

Supplementary Data 1 from ref. 40 was downloaded, and genes expressed in both the 6 hpf dataset from Lee et al. and the 5.5 hpf dataset analyzed using the mCPBA/TFEA pH 7.4 method were identified. Genes classified as maternal, weakly maternal, and zygotic in the 6 hpf dataset from Lee et al. were compared with genes categorized as maternal, maternal-zygotic, and zygotic using the mCPBA/TFEA pH 7.4 method. A Fisher's exact test was performed to assess the statistical significance of gene set overlaps. To further illustrate the shared zygotic genes between these classifications, a Venn diagram was generated (Supplementary Fig. 9a).

### Chk1 data processing

Raw 4 hpf zebrafish embryo sequencing data corresponding to Chk1 overexpression and Triptolide-treated Chk1 overexpression conditions were obtained from the study by ref. 42 (NCBI Sequence Read Archive, accession numbers SRR8552566 and SRR8552569). The raw FASTQ files were subjected to quality control, where low-quality bases and adapter sequences were trimmed and filtered using fastp with default parameters.

The processed reads were then mapped to the *Danio rerio* reference genome (GRCz11, Ensembl release 108) using HISAT2 v2.2.1[65] with default settings. Properly aligned reads were assigned to genomic features for gene-level quantification using featureCounts v2.0.3[66] with parameters -p and -t exon, ensuring that only exonic regions were considered for read aggregation at the gene level. The resulting gene expression matrix was used for downstream analysis.

Genes with a total count greater than 10 across all samples were retained for downstream analysis. The filtered count matrix was then normalized using the reads per kilobase per million mapped reads (RPKM) method. The log2 fold change (log2FC) values were computed as the logarithm (base 2) of the ratio between the RPKM-normalized expression levels in the Chk1 overexpression and Triptolide-treated Chk1 overexpression conditions. Genes with log2 fold change values greater than or equal to 2 were selected for further analysis. To assess the overlap between differentially expressed genes and the target genes identified using the mCPBA/TFEA pH 7.4 method, a Venn diagram was constructed to visualize the common and unique gene sets (Supplementary Fig. 9e). Significant cell type marker genes from the 5.5 hpf embryo dataset were selected based on an adjusted $p$-value threshold of less than 0.05. Among these, the top 20 overlapping genes with the differentially expressed gene set were identified. To visualize their expression patterns, violin plots were generated, illustrating the distribution of expression levels across different cell type (Supplementary Fig. 9g). Then, based on the log2 fold change (log2FC) values of the commonly detected genes and their NTR values derived from the mCPBA/TFEA pH 7.4 method, a scatter plot was generated to illustrate the distribution of these genes. This visualization provided insights into the relationship between differential expression patterns and transcript stability across conditions, allowing for a comparative assessment of gene regulation dynamics (Supplementary Fig. 9f).

### 5.5 hpf embryo dataset cell cycle analysis

For the 5.5 hpf embryo chemical perturbation dataset, cell cycle analysis was performed using the CellCycleScoring function in Seurat to classify cells into G1, S, and G2M phases. The scoring genes were derived from the default cell cycle marker set in Seurat, with homologous zebrafish genes identified using Ensembl Biomart. To explore the proportion of different cell cycle phases within various cell types, an alluvial plot was generated using ggalluvial[67] and ggplot2 (Supplementary Fig. 9c). Additionally, based on NTR values, maternal genes (<5%) and zygotic genes (>70%) were identified within each cell cycle phase. The G1-phase cells in primordial germ cells (PGCs) were too few in number to accurately determine the distribution of maternal-zygotic genes. A bar plot was generated to illustrate the proportions of

maternal, maternal-zygotic, and zygotic genes in different cell types across cell cycle phases (Supplementary Fig. 9d).

### Statistics & reproducibility

No statistical method was used to predetermine sample size. No data were excluded from the analyses. The experiments were not randomized. The investigators were not blinded to allocation during experiments and outcome assessment.

### Data visualization

All plots were generated using the ggplot2 v3.4.4 (https://github.com/tidyverse/ggplot2), cowplot v1.1.1 (https://github.com/wilkelab/cowplot), ComplexHeatmap v2.14.0[68] and matplotlib v3.6.3 (https://github.com/matplotlib/matplotlib) packages. Box plots display the median (center line) and interquartile range (25th to 75th percentile), with whiskers representing 1.5 times the interquartile range, and circles indicating outliers. Violin plots include a gray line on each side, representing a kernel density estimate of the data distribution. Wider sections indicate higher probability, while thinner sections indicate lower probability.

### Reporting summary

Further information on research design is available in the Nature Portfolio Reporting Summary linked to this article.

## Data availability

All sequencing data associated with this study are available in the Genome Sequence Archive at the National Genomics Data Center under accession no. CRA023644. Source data are provided with this paper. Published datasets used in this study are available in the NCBI Gene Expression Omnibus, under accessions GSE158849, GSE224918, GSE47558, and SRP184786 (for the zygotically expressed genes comparison). Source data are provided with this paper.

## Code availability

The analysis source code is available on the GitHub repository (https://github.com/penghu-sc/Benchmarking-Metabolic-RNA-Labeling/) and source code has been deposited in Zenodo (https://doi.org/10.5281/zenodo.15567719).

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

## Acknowledgements

We thank Dr. Qi Qiu and Prof. Hao Wu for their valuable comments, which significantly improved the manuscript and provided important insights into the experimental design. We also thank Dr. Mudan He and Prof. Yonghua Sun for their valuable suggestions on whole-mount in situ hybridization of zebrafish embryos. We appreciate Prof. Xi Chen and Prof. Ni Hong for their insightful feedback during the revision of this paper. We are grateful to the members of the Hu lab for their valuable discussions. The schematic illustrations in Figs. 3, 4 and Supplementary Fig. 1 were created or modified using images from BioRender.com. The zebrafish resources used in this study were provided by the China Zebrafish Resource Center, National Aquatic Biological Resource Center, CZRC/NABRC. We thank Instrumental Analysis Center of Shanghai Jiao Tong University for helping with osmium tetroxide chemical condition. We thank the computational resources and support provided by Modern Information and Educational Technology Center of Shanghai Ocean University. This work was supported by the National Natural Science Foundation of China (Grants No. 32200414, No. 32373113, and No. 32341061 to P.H., and No. 32400418 to W.L.) and SciTech Funding by CSPFTZ Lingang Special Area Marine Biomedical Innovation Platform.

## Author contributions

X. Zhang and P.H. conceived the study. X. Zhang and M.P. conducted most of the experiments with the help from J.Z. and P.H. M.P. and P.H. performed computational analysis. X. Zhai, C.W., and H.J. assisted with scRNA-Seq library preparation. Z.W. and Q.X. contributed to zebrafish maintenance. S.H., M.L., and W.L. contributed to data analysis and visualization. W.Y. assisted with zebrafish microinjection. K.M. and L.C. contributed to the experimental design. X. Zhang, M.P., and P.H.

analyzed the results and wrote the manuscript with inputs from all authors. P.H. supervised the study.

## Competing interests

The authors declare no competing interests.
