## [Transparent Peer Review file · Nature Communications]

Benchmarking metabolic RNA labeling techniques for high-throughput single-cell RNA sequencing

Corresponding Author: Professor Peng Hu

Version 0:

Reviewer comments:

Reviewer #1

(Remarks to the Author)

Understanding the dynamic regulation of gene expression at the single cell level requires robust methods to detect not just RNA levels, but also their changes over time. In their manuscript, "Benchmarking Metabolic RNA Labeling Techniques for High-Throughput Single-Cell RNA Sequencing," Zhang and co-workers provide a rigorous comparison of methodologies to address this challenge. The authors provide a well-designed set of comparisons using several different chemically approaches to detect metabolically labeled transcripts and two different approaches to capture RNA from single cells. The authors made many excellent choices to ensure their comparisons within their new data and with data from literature were fairly conducted. They found that the optimal chemistry for detecting metabolically labeled transcripts depends on the platform, especially because of the chemical compatibility of different platforms. The authors demonstrated the applicability of the methodology with a nice study on the RNA dynamics during the maternal to zygotic transition in zebra fish. The authors conclusions are balanced, nuanced, and well supported by their data and analyzes. This is an important paper that helps establish a better methodological foundation for an exciting field.

Minor comments:

- 1) In Fig 2a it is very difficult to parse which sample is which. Perhaps the authors could switch to plotting by category like in Fig 2b and make the GT, GC, etc different colors, or find some other way to make it easier to figure out which method corresponds to which bar.
- 2) I do not understand what is supposed to be learned from Fig 2f. Why would treatment-specific clusters show up in UMAP and what does their absence say? Both the text and what was supposed to be learned from the figure were unclear to me. I suspect this analysis is to address the possibility that some of the methods might be biased for detection of different cell states with different transcriptome dynamics, but if so this could be better explained.

Reviewer #2

(Remarks to the Author)

Time-resolved scRNA-seq using metabolic RNA labeling is an advanced technology to capture the RNA dynamics in complex biological processes with single-cell resolution. In this manuscript, Zhang et al. performed benchmarking of seven chemical conversion methods for metabolic RNA labeling using the Drop-seq platform. They found an optimal method and applied the method to profile the single-cell RNA dynamics of zebrafish embryonic cells during the maternal-to-zygotic transition. They also evaluated a commercial platform MGI C4, which is similar to Drop-seq and compared the compatibility of on-bead IAA treatment with Drop-seq platform.

Overall, the benchmarking is important for the rapidly growing field of metabolic RNA labeling-based scRNA-seq. A guidance for selecting the most appropriate chemical conversion method and scRNA-seq platform will boost the broader application of the technology for biological research. However, the manuscript by Zhang et al. did not provide a comprehensive benchmarking as expected. The following issues need to be resolved.

Major issues:

1. The performance of metabolic RNA labeling-based scRNA-seq relies not only on the selected chemical conversion methods but also on the selected scRNA-seq platforms. In this manuscript, the authors only compared the performance of chemical conversion methods on droplet microfluidics-based scRNA-seq platforms (both Drop-seq and MGI C4). Other scRNA-seq platform used in metabolic RNA labeling based high-throughput scRNA-seq were ignored. Split-and-pool-based

sci-RNA-seq in sci-fate, microwell-based Well-paired-seq in Well-TEMP-seq, and 10x Chromium-based scRNA-seq in scSLAM-seq should be added to the benchmarking for systematic comparison.

2. The seven tested chemical conversion methods did not include the TUC-seq chemistry, which involves osmium tetroxide (OsO₄) and ammonium chloride (NH₄Cl).

3. The sequencing depth affect the number of detected genes, which may lead to different biological observations such as the discovery of novel zygotic genes in zebrafish embryogenesis. Detailed evaluation and discussion should be provided to justify the identification of novel zygotically activated transcripts in zebrafish embryogenesis.

4. The optimal chemical conversion method applied in the maternal-to-zygotic transition in zebrafish did not generate new and solid findings. More profound applications of the optimized metabolic RNA labeling-based scRNA-seq are necessary to demonstrate the importance of the benchmarking and optimization.

Minor issues:

1. In line 98, the description of “metabolic labeling (200 μM 4sU) for 4 hours” is not consistent with the conditions described in Methods. The concentration of 4sU may be 100 μM.

2. The result of at least one negative control gene is needed in Fig. 3g.

Reviewer #3

(Remarks to the Author)

In this manuscript, Zhang et al. performed single-cell sequencing on RNAs the metabolically labelled by 4-thiouridine (4-sU) in both cell culture and early zebrafish embryos. By comparing seven different conditions (plus one commercial platform) for the chemically induced base conversions, they found striking difference in the efficiency of base conversions and highlighted that some methods (such as those use mCPBA) outperformed others and that the commercial platform worked best when cell numbers are small. In addition, using those validated approaches with better performance, they discovered hundreds of new zygotic genes activated in early zebrafish embryo development, including some cell cycle genes - those genes were unable to be picked up by previous studies. Thus, this work represents highly technical improvement, which has important values in revealing new biological insights. Because single-cell sequencing approaches are increasingly important for measuring novel gene expression patterns and characterizing new cellular states or types - particularly for detecting nascent transcripts in those critical transitional biological processes such as gene activation in early development - this work will be of great interest to a large number of readers this journal and thus worth publishing. However, there are several caveats of this manuscript that needs to be improved prior to its publication, and it would be great if the authors could address the following points.

Major points:

1. This work is technically heavy, but the authors failed to put a similar weight on advancing the new biological insights that could be revealed by these data-rich scRNA-seq, which would weaken the importance of this work and reduce the enthusiasm of applying these approaches. The work could be significantly improved by providing more careful analysis of the data that could lead to novel biological insights. For example, the authors found that as compared with the housekeeping genes, the cell cycle genes have a faster turnover rate and shorter half-lives in a cell line. This claim has apparently been well known and not novel. Further, it is not surprising that higher expression was detected when the base conversion was increased. Instead, because typically the cells spend quite some time in the S phase, it is surprising that the authors were able to only obtain a very low representation of the cycling cells (644 out of 41,240 cells, ~ 1.5%). Did the authors verify the cell cycle states of the bulk cells (e.g., by flow cytometry) and whether the capture of the cycle states during scRNA-seq was consistent with that of bulk analysis? This is important because some cells or RNAs could be lost during the process of scRNA-seq library prep or chemical conversion reactions, which could bias towards the cycling cells. In addition, it may be beyond the scope of this work, but it would be nice if the authors could validate the mechanisms of the new cell cycle gene expression from the scRNA-seq datasets by manipulating the cells, e.g., using specific cell cycle inhibitors.

2. A second major biological insight provided by the authors is classification of the three categories of genes (M, Z and MZ) and identification of new zygotically activated genes during zebrafish early embryogenesis. Interestingly, the authors discovered that ~ 76-80% genes with new transcription are MZ genes (Fig. 3e). This finding is very consistent with that analyzed by the bulk RNA-seq in zebrafish (e.g., Antonio Giraldez group showed 74% genes activated during ZGA are MZ genes; see Lee et al., Nature 2014) and by other nascent RNA-seq approaches in other species such as *Xenopus* (e.g., Matthew Good group showed 78.4% genes with nascent transcription during ZGA are MZ genes; see Chen et al., Curr Biol 2022). The authors should discuss this important finding, which helps validate the single-cell seq approaches presented by the authors. Further, there is still a gap in understanding the mechanisms underlying the significant maternal contribution to new transcription in early phases of development. Recently, work in various model embryonic systems has suggested a critical role of cell cycle in activating gene expression during ZGA. Since the authors were able to detect the cell cycle states in the zebrafish cell line as mentioned above, did they look at whether the cell cycle length affects nascent transcription in early zebrafish embryos at the single-cell level? It would be wonderful if the author could include those data analyses to support this very important mechanism, if not providing additional mechanistic insights by manipulating the embryos with specific cell cycle inhibitors.

3. This manuscript is exclusively focused on the metabolic labeling as a result of chemical-induced conversion of bases such as using the nucleoside analog 4-SU, which is indeed powerful and has wide applications. However, there are many other metabolic labeling approaches to profile nascent RNA transcripts in single cells, which are not mentioned or discussed in depth by the authors. For example, recently, using the click chemistry-based approaches, Philip Sharp group developed scGRO-seq (Mahat et al., Nature 2024) to sequence the single-cell nascent RNAs, which revealed new mechanisms of coordinated transcription. While it is beyond the scope of the current work to experimentally compare the efficiency of detecting nascent transcripts metabolically labeled by differential modified bases in single cells, which is of course both technically and biologically important, the authors should at least cite those works and provide sufficient justification and discussion of why the chemical-induced conversion methods they focused on are superior than other metabolic labeling approaches - this would not only help contextualize this work but also magnify its significance.

4. This study is specifically focused on profiling the poly(A)-tailed mRNA; however, as many non-coding RNAs, such as microRNAs, lincRNAs and repetitive endogenous retrovirus elements, that play essential roles in early embryogenesis, are activated during the zygotic genome activation. To generalize its application and draw solid conclusions, it would be important for the authors to extend this work to those non-mRNAs too as these scRNA-seq approach could be invaluable in revealing novel insights into gene regulation. We do not request the authors to perform additional experiments, but they should point out the limitations of this work somewhere in the manuscript.

5. It is not surprising that the commercial MGI C4 outperforms other home-brew approaches, particularly when the cell numbers are rare which is often more than a concern for certain types of tissues such as early embryos. This conclusion overshadowed all other approaches presented by the authors, which is disappointing. That said, the hard work presented by the authors is still valuable, which serves a guidance for future method selection. In the decision tree in Fig. 5, are there any other factors should be considered when choosing the methods and platforms (such as cost and cell types)?

Minor points:

1. Fig. 1 and Fig. 5 are helpful illustrations, but neither suffices a main figure. It is recommended to move them to the supplemental data.
2. Several figures (Fig 2c, 2d, 2f, 2g; Fig 4e, 4f) are hardly readable. The presentation of these figures should be improved by simplifying the plots.
3. Negative controls and statistics for the ISH by each gene should be included in Fig. 3g.
4. It would be nice if the author could provide some rationale for using the ZF4 cell line for their "benchmark" analysis.
5. For defining the cycling state, what data are used in Extended Data Fig. 2? More specific and detailed info how the authors processed them should be provided in the Methods and/or the legend.
6. The ambiguous expression should be clarified in Line 138: "This suggests that whether the chemical conversion is conducted in intact cells (in-situ) or after mRNA release (on-beads) introduces greater transcriptomic variation than the specific chemical conversion methods employed (Extended Data Fig. 2b)."
7. References should be provided for the following:
 - (1) Line 39: "Metabolic RNA labeling combined with scRNA-seq has significantly enhanced our ability to quantitatively analyze RNA synthesis and degradation, leading to key discoveries." Also, what "key discoveries" do the authors mean?
 - (2) Line 158: "Additionally, the top 10% most stable and unstable transcripts were enriched for GO terms similar to those found in mouse embryonic stem cells (Fig. 2h)."
 - (3) Line 268: "The results showed that mCPBA/TFEA-based methods maintained their efficiency, achieving T-to-C substitution rates comparable to, or slightly higher than, those observed in published datasets using fresh cells."
8. The expression of "fair comparison" should be avoided due to its subjectivity. It should be "direct comparison".
9. Some typos and grammatic errors should be corrected, including but not limited to:
 - (1) Line 42: "investigation of" should be "investigating".
 - (2) Line 141: "Some maker genes" should be "Some marker genes".
 - (3) Line 179: "and another" should be "and the other".
 - (4) Line 289: "mutually" should be "multiple".

Version 1:

Reviewer comments:

Reviewer #2

(Remarks to the Author)

The revised manuscript has addressed the reviewers' concerns and it's recommended for publication in the journal.

Reviewer #3

(Remarks to the Author)

In this revision, Zhang et al. have performed additional experiments and analyses as suggested, edited the texts, cited relevant literature, clarifies the ambiguities, and made the study more comprehensive and concise in presentation. We believe that the authors have addressed all our previous comments, and this work has been significantly improved. This study provides an important guideline in the field of analyzing nascent transcription in single cells, and it would be of great interest to the readers. Thus, we recommend that this revision is ready for publishing.

Our point-by-point responses to their comments are listed below:

Reviewers' comments:

Reviewer #1 (Remarks to the Author):

Understanding the dynamic regulation of gene expression at the single cell level requires robust methods to detect not just RNA levels, but also their changes over time. In their manuscript, "Benchmarking Metabolic RNA Labeling Techniques for High-Throughput Single-Cell RNA Sequencing," Zhang and co-workers provide a rigorous comparison of methodologies to address this challenge. The authors provide a well-designed set of comparisons using several different chemically approaches to detect metabolically labeled transcripts and two different approaches to capture RNA from single cells. The authors made many excellent choices to ensure their comparisons within their new data and with data from literature were fairly conducted. They found that the optimal chemistry for detecting metabolically labeled transcripts depends on the platform, especially because of the chemical compatibility of different platforms. The authors demonstrated the applicability of the methodology with a nice study on the RNA dynamics during the maternal to zygotic transition in zebra fish. The authors conclusions are balanced, nuanced, and well supported by their data and analyzes. This is an important paper that helps establish a better methodological foundation for an exciting field.

Response 1: We sincerely appreciate the reviewer's positive feedback on our manuscript.

Minor comments:

1) In Fig 2a it is very difficult to parse which sample is which. Perhaps the authors could switch to plotting by category like in Fig 2b and make the GT, GC, etc different colors, or find some other way to make it easier to figure out which method corresponds to which bar.

Response 2: We appreciate the reviewer's insightful suggestion to improve the clarity of Fig. 2a. In response, we have revised the figure to display only **T-to-C substitution rates**, as this is the primary focus of the chemical conversion efficiency analysis. Additionally, we have included **three additional chemical conditions**—adopted from **Well-Temp-Seq, sci-fate, and OsO₄-based reactions**—to provide a more comprehensive comparison. To further enhance readability, the original **Fig. 2a**, which included all mutation types (GT, GC, etc.), has been **relocated to the supplementary materials** (now **Supplementary Fig. 2a**) for reference. This restructuring allows for a clearer and more intuitive presentation of the key findings while retaining access to the full dataset. We hope this revision addresses the reviewer's concern and improves the overall clarity of the figure.

Revised Fig. 2a. Box plot showing T-to-C substitution rates across ten chemical conversion methods in 4sU-labeled ZF4 cells.

2) I do not understand what is supposed to be learned from Fig 2f. Why would treatment-specific clusters show up in UMAP and what does their absence say? Both the text and what was supposed to be learned from the figure were unclear to me. I suspect this analysis is to address the possibility that some of the methods might be biased for detection of different cell states with different transcriptome dynamics, but if so this could be better explained.

Response 3: We appreciate the reviewer's insightful comments and the opportunity to clarify.

1) To assess whether different chemical conversion methods affect the detection of specific cell states, particularly actively cycling cells, we performed additional analyses. Our results (now included in **Supplementary Fig. 3b,c**) revealed variations in the proportion of cycling cells across chemical conditions. Compared to the control (~1% dividing cells), chemical treatments altered cell-type composition, ranging from 0.04% to 3.30%, suggesting that certain reactions may introduce biases in detecting specific cell populations. Notably, OsO4 and IAA (37°C) treatments almost completely depleted cluster 1, which consists of dividing cells expressing high levels of mitotic markers (**Supplementary Fig. 3a**). This suggests that these conditions may selectively impair the detection of mitotic cells, likely due to their impact on RNA stability or conversion efficiency in highly transcriptionally active cells. Additionally, the low proportion of dividing cells (~3% in ZF4) may further hinder accurate quantification (**Lines 341-345 in the discussion part of revised manuscript**)

Supplementary Fig. 3

Revised Supplementary Fig. 3 | The Proportion of Different Cell States. **a.** Violin plots showing the expression of mitotic genes in different cluster. **b-c.** UMAP visualization showing datasets respectively from control and the ten chemical conversion methods. **c.** Box plot showing dividing cluster proportion across control and the ten chemical conversion methods in 4sU-labeled ZF4 cells.

2) To ensure that 4sU labeling itself does not alter cell cycle dynamics, we performed flow cytometry analysis,

which confirmed that the addition of 4sU did not affect cell cycle distribution. The proportion of G2/M phase cells remained around 3% (**Supplementary Fig. 3d**).

Revised Supplementary Fig. 3d. Flow cytometry analysis of cell cycle distribution of ZF4 cells. The cells were treated without (left three) or with 4sU (right three).

We have revised the main text (**Lines 159-165**) to clarify these findings and their implications. We hope these updates address the reviewer's concerns and improve the manuscript's clarity.

Reviewer #2 (Remarks to the Author):

Time-resolved scRNA-seq using metabolic RNA labeling is an advanced technology to capture the RNA dynamics in complex biological processes with single-cell resolution. In this manuscript, Zhang et al. performed benchmarking of seven chemical conversion methods for metabolic RNA labeling using the Drop-seq platform. They found an optimal method and applied the method to profile the single-cell RNA dynamics of zebrafish embryonic cells during the maternal-to-zygotic transition. They also evaluated a commercial platform MGI C4, which is similar to Drop-seq and compared the compatibility of on-bead IAA treatment with Drop-seq platform. Overall, the benchmarking is important for the rapidly growing field of metabolic RNA labeling-based scRNA-seq. A guidance for selecting the most appropriate chemical conversion method and scRNA-seq platform will boost the broader application of the technology for biological research. However, the manuscript by Zhang et al. did not provide a comprehensive benchmarking as expected. The following issues need to be resolved.

Response 4: We sincerely appreciate the reviewer's positive feedback on our benchmarking work.

Major issues:

1. The performance of metabolic RNA labeling-based scRNA-seq relies not only on the selected chemical conversion methods but also on the selected scRNA-seq platforms. In this manuscript, the authors only compared the performance of chemical conversion methods on droplet microfluidics-based scRNA-seq platforms (both Drop-seq and MGI C4). Other scRNA-seq platform used in metabolic RNA labeling based high-throughput scRNA-seq were ignored. Split-and-pool-based sci-RNA-seq in sci-fate, microwell-based Well-paired-seq in Well-TEMP-seq, and 10x Chromium-based scRNA-seq in scSLAM-seq should be added to the benchmarking for systematic comparison.

Response 5: We sincerely thank the reviewer for highlighting this point and acknowledging the importance of comparing various single-cell RNA-seq platforms. While our initial benchmarking focused on droplet microfluidics-based scRNA-seq platforms (Drop-seq and MGI C4), we recognize that other platforms, such as sci-RNA-seq in sci-fate^{1,2,3} and Well-Paired-seq in Well-TEMP-seq⁴⁻⁶, also play a crucial role in metabolic RNA labeling-based single-cell transcriptomics. In response to the reviewer's suggestion, we have expanded our study through additional experiments and analyses to enhance its comprehensiveness.

1) We have provided a more comprehensive summary of metabolic RNA labeling-based scRNA-seq technologies, detailing their distinct features and applications. This summary has been incorporated into **Supplementary Fig. 1 (referenced in Lines 64-77, 92, 98-102, 109-110 in the revised manuscript)** to ensure our study reflects the breadth and advancements in this field.

Supplementary Fig. 1

Revised Supplementary Fig. 1. Overview of currently available 4sU-labeling single-cell genomics technologies.

2) Our study primarily aims to benchmark different chemical conversion methods and assess their compatibility with scRNA-seq platforms. To ensure a direct and unbiased comparison, we focused on a single platform, the home-brew Drop-seq system⁷, due to its flexibility and suitability for experimental modifications. In **Supplementary Fig. 1**, we now illustrate the classification of 10 chemical labeling strategies, categorized as either in-situ reactions (e.g., 10x Genomics^{8,9}, sci-fate¹/ sci-fate2²) or on-bead reactions (e.g., scNT-seq¹⁰ and Well-TEMP-seq^{4,5}). We have now included three additional chemical conditions from sci-fate/sci-fate2, Well-TEMP-seq, and TUC-seq¹¹, bringing the total number of evaluated chemical reactions to ten (**Fig. 1b**). To further assess the efficiency of these 10 chemical reactions, we analyzed T-to-C substitution rates, new RNA fractions, and library complexity (genes and UMIs per cell). These results have been incorporated into revised **Fig. 2a-c** and **Supplementary Fig. 2a-c** for a more detailed evaluation.

b Chemical conversion methods

Method	Reagent	Buffer pH	Temperature	Time	Reference	
SLAM -based	In-situ (In fixed cells)	IAA	7.4	50 °C	15 mins	Herzog et al., 2017 (SLAM-seq); Holler et al., 2021.
	In-situ (In fixed cells)	IAA	8.0	50 °C	15 mins	Herzog et al., 2017 (SLAM-seq); Cao et al., 2020 (Sci-fate); Maizels et al., 2024 (Sci-fate2).
	On-beads	IAA	8.0	32 °C	15 mins	Fishman et al., 2024 (scSLAM-seq).
	On-beads	IAA	8.0	37 °C	~1 hr	Lin et al., 2023 (Well-TEMP-seq); Yin et al., 2024 (Well-TEMP-seq).
TimeLapse -based	On-beads	mCPBA+TFEA	5.2	45 °C	~1.5 hrs	Schofield et al., 2018 (TimeLapse-seq); Zimmer et al., 2021 (STL-seq); Zimmer et al., 2023.
		mCPBA+TFEA	7.4	45 °C	~1.5 hrs	Zimmer et al., 2023.
		NaIO ₄ +TFEA	5.2	45 °C	~1.5 hrs	Schofield et al., 2018 (TimeLapse-seq); Qiu et al., 2020 (scNT-seq); Zimmer et al., 2023.
		NaIO ₄ +TFEA	7.4	45 °C	~1.5 hrs	Zimmer et al., 2023.
		NaIO ₄ +NH ₄ Cl	8.8	45 °C	~1.5 hrs	Zimmer et al., 2023.
TUC-based	On-beads	OsO ₄ +NH ₄ Cl	8.8	50 °C	~3 hrs	Riml et al., 2017 (TUC-seq); Zimmer et al., 2023.

Revised Fig. 1b. Summary of the ten chemical conversion methods evaluated in this study, including key parameters such as the main reagent, buffer pH, temperature, reaction time, and relevant references. "In-situ" refers to chemical conversion occurring within intactly fixed cells, while "on-beads" indicates that the chemical conversion occurs after mRNA is released from the cells and captured on beads.

Revised Fig. 2a-c | a. Box plot showing T-to-C substitution rates across ten chemical conversion methods in 4sU-labeled ZF4 cells. **b-c.** Scatterplots comparing the number of genes (c) or UMIs (d) detected per cell as a function of aligned reads per cell across the ten chemical conversion methods. Color indicates treatment methods.

Supplementary Fig. 2

Revised Fig. 2a-c | **a.** Box plot showing nucleotide substitution rates across ten chemical conversion methods in 4sU-labeled ZF4 cells. **b.** Violin plots showing the distribution of labeled RNA fractions across the ten chemical conversion methods. **c.** Proportion of UMIs containing T-to-C substitutions under ten chemical conversion methods.

3) Given the unique features and workflows of different single-cell platforms, direct benchmarking between methods like sci-fate and Well-TEMP-seq requires specialized experimental setups and customized protocols. Notably, Well-TEMP-seq uses Drop-seq beads, similar to scNT-seq, making it compatible with on-bead chemical conversions. To broaden platform-level comparisons, we conducted additional benchmarking across 10x Genomics, MGI C4, and home-brew Drop-seq, leading to the following key findings (**revised Fig. 4**): Our key findings including: On-bead chemical conversion with MGI C4 outperformed other methods in terms of T-to-C mutation rates and library complexity. Due to the incompatibility of on-bead reactions with 10x Genomics, we applied in-situ chemical conversion instead (In-situ, pH 8.0). Results showed that 10x Genomics (nGene: 670, nUMI: 1,745) and MGI C4 (nGene: 624, nUMI: 1,263) exhibited comparable library complexity at 4k reads per cell. Within the same platform, on-bead chemical reactions (On-beads IAA, 32°C) consistently outperformed in-situ reactions, whether using home-brew Drop-seq or MGI C4 (**revised Fig. 4c-f**).

Revised Fig. 4c-f | Comparative analysis of single-cell platforms and chemical conversion methods. c. Proportion of UMIs containing T-to-C substitutions under different conditions and platforms. The color gradient indicates the number of T-to-C substitutions per read, with darker shades representing a higher number of substitutions within the UMI. "Ctrl" represents the control group without chemical treatment; "In-situ" refers to "In-situ IAA, pH8.0" method, while "on-beads" indicates "On-beads IAA, 32°C" chemistry in **c-f**. **d.** Box plot showing T-to-C substitution rates across the scRNA-seq platforms. Different colored boxes represent various platforms and treatment methods. **e-f.** Scatterplots showing the number of genes (e) or UMIs (f) detected per cell as a function of aligned reads per cell across the different platforms.

While we appreciate the reviewer's suggestion to include broader platform comparisons, conducting all suggested experiments within a reasonable timeline is not feasible due to the extensive resources required. We believe the above significant additions substantially enhance the comprehensiveness of our study and effectively address the reviewer's concerns. We sincerely appreciate the opportunity to improve our manuscript and thank the reviewer for their valuable feedback.

2. The seven tested chemical conversion methods did not include the TUC-seq chemistry, which involves osmium tetroxide (OsO₄) and ammonium chloride (NH₄Cl).

Response 6 : We thank the reviewer for raising this point. We have now included the OsO₄/NH₄Cl reaction comparison in the **revised Fig. 2a-c**. As previously reported¹², the OsO₄/NH₄Cl reaction demonstrates inferior performance compared to the chemical conversion methods already included in our study (mCPBA/TFEA pH 5.2, NaIO₄/TFEA pH 5.2, NaIO₄/TFEA pH 7.4, and NaIO₄/NH₄Cl pH 8.8). This comparison further supports our rationale for prioritizing high-efficiency chemical reactions in our benchmarking analysis (**Lines 139-151 in the revised manuscript**).

Revised Fig. 2a-c | **a**. Box plot showing T-to-C substitution rates across ten chemical conversion methods in 4sU-labeled ZF4 cells. **b-c**. Scatterplots comparing the number of genes (c) or UMIs (d) detected per cell as a function of aligned reads per cell across the ten chemical conversion methods. Color indicates treatment methods.

3. The sequencing depth affect the number of detected genes, which may lead to different biological observations such as the discovery of novel zygotic genes in zebrafish embryogenesis. Detailed evaluation and discussion should be provided to justify the identification of novel zygotically activated transcripts in zebrafish embryogenesis.

Response 7: We thank the reviewer for bringing this to our attention. We agree that sequencing depth affects the number of detected genes, which is why we benchmarked the methods by comparing library complexity in terms of detected genes and UMIs at the same sequencing depth to ensure a fair comparison (as shown in **Fig. 2** and **Fig. 4**). We also acknowledge the importance of further evaluating and validating our identified novel zygotic-activated transcripts. To address this, we have conducted additional analyses and experiments as outlined below:

1) Comparison with Published Studies

To validate our zebrafish findings, we compared our results with previously published datasets. In Fig. 3e, we found that ~76–80% of genes exhibiting new transcription were classified as MZ genes. This observation aligns closely with prior studies using bulk RNA-seq in zebrafish. For example, a study by Lee et al. from the Antonio Giraldez group reported that 74% of genes activated during ZGA are MZ genes¹³. Similarly, nascent RNA-seq approaches in other species, such as *Xenopus*, revealed comparable results, with 78.4% of nascently transcribed genes classified as MZ genes (*Chen et al., Curr Biol, 2022*)¹⁴. These consistent findings support the reliability of our metabolic RNA labeling approach (**Lines 219-223 in the revised manuscript**).

2) Comparison with Previously Identified Zygotic Genes

To further validate our identified zygotic genes, we compared our dataset with the zygote-specific gene list from Lee et al., *Nature*, 2013, which was generated by capturing intron signals of zygotic transcripts¹³. The results showed a significant overlap between our 4sU-labeled zygotic genes and the previously identified zygotic gene set (odds ratio = 12.13, $p < 2.2e-16$, Fisher's exact test, **Supplementary Fig. 9a**). This strong overlap further confirms the robustness of our approach in identifying zygotic gene activation (**Lines 243-246 in the revised manuscript**).

Supplementary Fig. 9

Revised Supplementary Fig. 9a. Venn diagram showing the overlap of defined zygotic genes among mCPBA-based method (pH 7.4) and published studies¹³.

3) Experimental Validation via Whole-mount *In Situ* Hybridization

Based on the above comparisons, we identified several novel zygotic genes, including *apoeb*, *akap12b*, *marcks1b*, *cited4b*, and *pnrc2*, in addition to commonly identified zygotic genes such as *tbx16*. To validate these findings, we further designed probes targeting the intronic regions of these six genes and additionally included *marcksa* and *setdb1b*, which were exclusively identified using the mCPBA/TFEA method. Their unspliced transcript expression at 5.5 hpf was validated using control probes for comparison (**Supplementary Fig. 8c** and see Methods, section “Whole-mount in situ hybridization of zebrafish embryos”). The results confirmed that these genes are actively transcribed during ZGA, further supporting their classification as novel zygotic genes (**Lines 239-242 in the revised manuscript**).

Revised Supplementary Fig. 8c. Antisense (top) and sense-strand negative control (NTC, bottom) across introns for whole-mount in-situ hybridization staining validation of 5.5 hpf zebrafish embryos for mRNAs of zygotic genes indicated in Fig. 3f,g. Scale bar: 200 μ m.

Together, these analyses and experiments provide strong evidence that our metabolic labeling approach accurately identifies zygotic transcription events and reveals novel ZGA-associated genes.

4. The optimal chemical conversion method applied in the maternal-to-zygotic transition in zebrafish did not generate new and solid findings. More profound applications of the optimized metabolic RNA labeling-based scRNA-seq are necessary to demonstrate the importance of the benchmarking and optimization.

Response 8 : We sincerely thank the reviewer for their valuable comments and for emphasizing the importance of demonstrating the broader biological applications of our optimized metabolic RNA labeling-based scRNA-seq approach. To further validate our findings and highlight the significance of our benchmarking and optimization, we conducted additional analyses and experiments (see Response 7), summarized below:

1) We compared our zebrafish ZGA findings with previously published bulk RNA-seq and nascent RNA-seq datasets, showing strong consistency. Our identified zygotic genes significantly overlapped with those reported in previous studies ($p < 2.2e-16$, Fisher's exact test, **Supplementary Fig. 9a**).

2) To confirm our identified zygotic genes, we designed intron-targeting probes for novel zygotic genes and

validated their expression during ZGA through whole-mount in situ hybridization (WISH) (**Supplementary Fig. 8c**).

3) Beyond benchmarking chemical conversion methods, we investigated the role of cell cycle lengthening in ZGA, a key aspect of early embryonic development. Studies in frogs and zebrafish suggest that as rapid cell divisions slow, nascent transcription increases (Chen et al., *Curr Biol* 2019; Chan et al., *Dev Cell* 2019)^{14,15}. In zebrafish 5.5 hpf embryos, our analysis revealed that PGCs had fewer cells in G1 phase, with higher zygotic gene expression in S and G2/M phases. PGCs, which predominantly remain in S/G2M, exhibited lower zygotic transcription and retained maternal mRNAs, reflecting distinct transcriptional regulation. Consistent with previous findings, our results show that G1 emerges progressively during ZGA but is not fully established at its earliest stages, suggesting that ZGA varies across cell types and is influenced by cell cycle dynamics (**Supplementary Fig. 9c,d; Lines 252-255 in the revised manuscript**).

Revised Supplementary Fig. 9 | c. Stacked bar chart showing the proportions of cell phases (G1, S or G2M) across different cell types. Colors indicate different cell phases. **d.** Stacked bar chart showing the proportions of identified maternal (M), maternal-zygotic (MZ), and zygotic (Z) genes across different cell types. Colors indicate gene types.

4) Finally, we investigate the relationship between cell cycle length and ZGA using *Chk1* Overexpression database.

While previous studies have shown that cell cycle arrest enhances zygotic transcription, its direct role in ZGA timing remains debated. In frogs, artificial and natural cell cycle lengthening promote ZGA. However, in zebrafish, *Chk1* mRNA overexpression blocks the formation of replication origins, slows cell divisions during the mid-blastula transition (MBT), and extends the cell cycle—but does not affect ZGA timing¹⁵. To investigate how cell cycle lengthening affects zygotic gene expression, we analyzed gene expression changes in *Chk1*-overexpressing (*Chk1* OE) embryos and compared them to control triptolide-treated *Chk1* OE embryos. We found that ~38.8% of *Chk1* OE-induced genes overlapped with zygotic genes identified in our study, indicating that while ZGA timing is independent of cell cycle length, a prolonged cell cycle enhances transcriptional competence (revised **Supplementary Fig. 9e**). Using our single-cell dataset, we identified cell-type-specific expression patterns among *Chk1*-induced zygotic genes (revised **Supplementary Fig. 9g**). *grhl3* and *cldn* were enriched in the EVL. *lft2* and *tbxta* were highly expressed in endoderm and mesoderm clusters. These results suggest that while cell cycle lengthening does not shift the timing of ZGA, it enhances the transcriptional potential of the embryo, contributing to lineage-specific gene activation (**Lines 256-265 in the revised manuscript**).

Revised Supplementary Fig. 9 | e. Venn diagram showing the overlap of defined zygotic genes (from Fig. 3e) among mCPBA-based method (pH 7.4) and published studies¹⁵, highlighting both unique and shared genes. **g.** Violin plots displaying the genes related to cell cycle length in different cell types of zebrafish embryos. Different color represents different cell types in zebrafish embryos.

Taken together, our study provides both methodological advancements and novel biological insights. While ZGA timing in zebrafish appears independent of cell cycle length, we demonstrate that cell cycle lengthening enhances transcriptional competence, enabling the activation of lineage-specific genes. By integrating single-cell metabolic labeling data with previous Chk1 OE studies, our work offers a refined perspective on how cell cycle dynamics shape early embryonic gene expression.

Minor issues:

1. In line 98, the description of “metabolic labeling (200 μ M 4sU) for 4 hours” is not consistent with the conditions described in Methods. The concentration of 4sU may be 100 μ M.

Response 9 : We thank the reviewer for pointing this out. We have corrected this typo in the revised manuscript to accurately reflect the concentration as 100 μ M (**Line 116 in the revised manuscript**).

2. The result of at least one negative control gene is needed in Fig. 3g.

Response 10 : We thank the reviewer for the comment. We have included the negative control for all the 8

genes in the revised manuscript to strengthen the validity of our findings (Supplementary Fig. 8c; Lines 239-242 in the revised manuscript).

Revised Supplementary Fig. 8c. Antisense (top) and sense-strand negative control (NTC, bottom) across introns for whole-mount in-situ hybridization staining validation of 5.5 hpf zebrafish embryos for mRNAs of zygotic genes indicated in Fig. 3f,g. Scale bar: 200 μ m.

Reviewer #3 (Remarks to the Author):

In this manuscript, Zhang et al. performed single-cell sequencing on RNAs the metabolically labelled by 4-thiouridine (4-sU) in both cell culture and early zebrafish embryos. By comparing seven different conditions (plus one commercial platform) for the chemically induced base conversions, they found striking difference in the efficiency of base conversions and highlighted that some methods (such as those use mCPBA) outperformed others and that the commercial platform worked best when cell numbers are small. In addition, using those validated approaches with better performance, they discovered hundreds of new zygotic genes activated in early zebrafish embryo development, including some cell cycle genes - those genes were unable to be picked up by previous studies. Thus, this work represents highly technical improvement, which has important values in revealing new biological insights. Because single-cell sequencing approaches are increasingly important for measuring novel gene expression patterns and characterizing new cellular states or types - particularly for detecting nascent transcripts in those critical transitional biological processes such as gene activation in early development - this work will be of great interest to a large number of readers this journal and thus worth publishing. However, there are several caveats of this manuscript that needs to be improved prior to its publication, and it would be great if the authors could address the following points.

Response 11 : We sincerely appreciate the reviewer's positive feedback on our manuscript.

Major points:

1. This work is technically heavy, but the authors failed to put a similar weight on advancing the new biological insights that could be revealed by these data-rich scRNA-seq, which would weaken the importance of this work and reduce the enthusiasm of applying these approaches. The work could be significantly improved by providing more careful analysis of the data that could lead to novel biological insights. For example, the authors found that as compared with the housekeeping genes, the cell cycle genes have a faster turnover rate and shorter half-lives in a cell line. This claim has apparently been well known and not novel. Further, it is not surprising that higher expression was detected when the base conversion was increased. Instead, because typically the cells spend quite some time in the S phase, it is surprising that the authors were able to only obtain a very low representation of the cycling cells (644 out of 41,240 cells, ~ 1.5%). Did the authors verify the cell cycle states of the bulk cells (e.g., by flow cytometry) and whether the capture of the cycle states during scRNA-seq was consistent with that of bulk analysis? This is important because some cells or RNAs could be lost during the process of scRNA-seq library prep or chemical conversion reactions, which could bias towards the cycling cells. In addition, it may be beyond the scope of this work, but it would be nice if the authors could validate the mechanisms of the new cell cycle gene expression from the scRNA-seq datasets by manipulating the cells, e.g., using specific cell cycle inhibitors.

Response 12 : We appreciate the reviewer's insightful suggestions and acknowledge the concerns regarding the interpretation of cell cycle states in our scRNA-seq dataset and potential biases introduced by chemical

conversion methods. In response, we conducted additional analyses and experiments to clarify these points. 1) To resolve ambiguity, we performed a detailed cell cycle marker analysis between Cluster 0 and Cluster 1 in our scRNA-seq dataset. Our findings show that genes associated with well-known mitotic cell cycle genes were significantly upregulated in Cluster 1 compared to Cluster 0. To reflect this distinction, we have renamed Cluster 1 as "dividing" rather than simply "cycling," as it specifically represents cells with mitosis (Revised **Supplementary Fig. 3a**; **Lines 157-159 in the revised manuscript**).

Revised Supplementary Fig. 3a. Violin plots showing the expression of mitotic genes in different cluster. “0” represent cluster “steady-state”; “1” represent cluster “dividing”.

2) We also observed method-dependent variation in detecting actively cycling cells. Notably, chemical conditions such as OsO₄ and IAA at 37°C led to an almost complete depletion of actively cycling cells, suggesting that certain chemical conversion methods may compromise the detection of cell cycle states (Revised **Supplementary Fig. 3b**). Additionally, the low proportion of dividing cells (~3% in ZF4) may further hinder accurate quantification (**Lines 341-345 in the discussion part of revised manuscript**).

Revised Supplementary Fig. 3b. Violin plots showing the expression of well-known mitotic cell cycle genes in different cluster.

3) To ensure that the low representation of dividing cells is accurate and to assess whether 4sU incorporation introduces biases, we compared cell cycle phase distributions between control and chemically treated conditions. Flow cytometry analysis using propidium iodide (PI) staining on ZF4 cells confirmed that 4sU labeling does not affect the detection of cycling cells (Revised **Supplementary Fig. 3d**). The proportion of G2/M phase cells remained around 3%, aligning with our clustering results. ZF4, a fibroblast cell line derived from zebrafish embryos, typically has a lower proportion of mitotic cells compared to mammalian cell lines such as 293T. To further validate this observation, we included 293T cells as a control in our experiment (see flow cytometry results below).

Revised Supplementary Fig. 3d. Flow cytometry analysis of cell cycle distribution of ZF4 cells. The cells were treated without (left three) or with 4sU (right three).

Flow cytometry analysis of cell cycle distribution of 293T cells.

We sincerely appreciate the reviewer's suggestion and agree that incorporating cell cycle inhibitors (e.g., CDK inhibitors, aphidicolin, triptolide) could further elucidate the relationship between cell cycle progression and new RNA synthesis. Future studies combining metabolic labeling scRNA-seq with targeted cell cycle perturbations will offer deeper insights into cell cycle-regulated gene expression dynamics and transcriptional bursting.

2. A second major biological insight provided by the authors is classification of the three categories of genes (M, Z and MZ) and identification of new zygotically activated genes during zebrafish early embryogenesis. Interestingly, the authors discovered that ~76-80% genes with new transcription are MZ genes (Fig. 3e). This finding is very consistent with that analyzed by the bulk RNA-seq in zebrafish (e.g., Antonio Giraldez group showed **74% genes** activated during ZGA are MZ genes; see Lee et al., Nature 2014) and by other nascent RNA-seq approaches in other species such as *Xenopus* (e.g., Matthew Good group showed **78.4% genes** with nascent transcription during ZGA are MZ genes; see Chen et al., Curr Biol 2022). The authors should discuss this important finding, which helps validate the single-cell seq approaches presented by the authors. Further, there is still a gap in understanding the mechanisms underlying the significant maternal contribution to new transcription in early phases of development. Recently, work in various model embryonic systems has suggested a critical role of **cell cycle** in activating gene expression during ZGA. Since the authors were able to detect the cell cycle states in the zebrafish cell line as mentioned above, did they look at **whether the cell cycle length affects nascent transcription** in early zebrafish embryos at the single-cell level? It would be wonderful if the author could include those data analyses to support this very important mechanism, if not providing additional mechanistic insights by manipulating the embryos with specific cell cycle inhibitors.

Response 13: We sincerely appreciate the reviewer's suggestion, which has greatly contributed to improving our results and discussion. In response, we have conducted additional analyses and experiments, as outlined below:

1) Comparison with Published Studies

To validate our zebrafish findings, we compared our results with previously published datasets. In Fig. 3e, we found that ~76–80% of genes exhibiting new transcription were classified as MZ genes. This observation aligns closely with prior studies using bulk RNA-seq in zebrafish. For example, a study by Lee et al. from the Antonio Giraldez group reported that 74% of genes activated during ZGA are MZ genes¹³. Similarly, nascent RNA-seq approaches in other species, such as *Xenopus*, revealed comparable results, with 78.4% of nascently transcribed genes classified as MZ genes (Chen et al., Curr Biol, 2022)¹⁴. These consistent findings support the reliability of our metabolic RNA labeling approach (Lines 219-223 in the revised manuscript).

2) Overlap with Previously Identified Zygotic Genes

To further validate our identified zygotic genes, we compared our dataset with the zygote-specific gene list from Lee et al. (*Nature*, 2013), which was generated by capturing intron signals of zygotic transcripts¹³. The results showed a significant overlap between our 4sU-labeled zygotic genes and the previously identified zygotic gene set (odds ratio = 12.13, $p < 2.2e-16$, Fisher's exact test, **Supplementary Fig. 9a**). This strong overlap further confirms the robustness of our approach in identifying zygotic gene activation (**Lines 243-246 in the revised manuscript**).

Supplementary Fig. 9

Revised Supplementary Fig. 9a. Venn diagram showing the overlap of defined zygotic genes among mCPBA-based method (pH 7.4) and published studies¹³.

3) Experimental Validation via Whole-mount *In Situ* Hybridization

Based on the above comparisons, we identified several novel zygotic genes, including *apoeb*, *akap12b*, *marcksl1b*, *cited4b*, and *pnrc2*, in addition to commonly identified zygotic genes such as *tbx16*. To validate these findings, we further designed probes targeting the intronic regions of these six genes and additionally included *marcksa* and *setdb1b*, which were exclusively identified using the mCPBA/TFEA method. Their unspliced transcript expression at 5.5 hpf was validated using control probes for comparison (**Supplementary Fig. 8c** and see Methods, section "Whole-mount in situ hybridization of zebrafish embryos"). The results confirmed that these genes are actively transcribed during ZGA, further supporting their classification as novel zygotic genes (**Lines 239-242 in the revised manuscript**).

Revised Supplementary Fig. 8c. Antisense (top) and sense-strand negative control (NTC, bottom) across introns for whole-mount in-situ hybridization staining validation of 5.5 hpf zebrafish embryos for mRNAs of zygotic genes indicated in Fig. 3f,g. Scale bar: 200 μ m.

4) Beyond benchmarking chemical conversion methods, we also investigated the role of cell cycle status in ZGA, a key aspect of early embryonic development. In zebrafish 5.5 hpf embryos, our analysis revealed that PGCs had fewer cells in G1 phase and exhibited lower zygotic transcription and retained maternal mRNAs, reflecting distinct transcriptional regulation. Consistent with previous findings^{14,15}, our results show that G1 emerges progressively during ZGA but is not fully established at its earliest stages, suggesting that ZGA varies across cell types and is influenced by cell cycle dynamics (**Revised Supplementary Fig. 9c,d, Lines 252-255 in the revised manuscript**).

Revised Supplementary Fig.9 | c. Stacked bar chart showing the proportions of cell phases (G1, S or G2M) across different cell types. Colors indicate different cell phases. **d.** Stacked bar chart showing the proportions of identified maternal (M), maternal-zygotic (MZ), and zygotic (Z) genes across different cell types. Colors indicate gene types.

5) We further investigate the relationship between cell cycle length and ZGA using *chk1* Overexpression database.

While cell cycle arrest can enhance zygotic transcription, its influence on cell-type-specific gene activation and the timing of ZGA remains under investigation. In frogs, cell cycle lengthening promotes ZGA¹⁴, but in zebrafish, *chk1* overexpression slows cell division without affecting ZGA¹⁵. To investigate how cell cycle lengthening affects zygotic gene expression, we analyzed gene expression changes in *Chk1*-overexpressing (*Chk1* OE) embryos and compared them to control triptolide-treated *Chk1* OE embryos. We found that ~38.8% of *Chk1* OE-induced genes overlapped with zygotic genes identified in our study, indicating that while ZGA timing is independent of cell cycle length, a prolonged cell cycle enhances transcriptional competence (revised **Supplementary Fig. 9e**). Using our single-cell dataset, we identified cell-type-specific expression patterns among *Chk1*-induced zygotic genes (revised **Supplementary Fig. 9g**). *grhl3* and *cldne* were enriched in the EVL. *lft2* and *tbxta* were highly expressed in endoderm and mesoderm clusters. These findings indicate that cell cycle lengthening primes lineage-specific gene activation, shaping early developmental transcriptional programs (**Lines 256-265 in the revised manuscript**).

Revised Supplementary Fig. 9 | e. Venn diagram showing the overlap of defined zygotic genes (from Fig. 3e) among mCPBA-based method (pH 7.4) and published studies¹⁵, highlighting both unique and shared genes. **g.** Violin plots displaying the genes related to cell cycle length in different cell types of zebrafish embryos. Different color represents different cell types in zebrafish embryos.

Taken together, these findings underscore the importance of our optimized metabolic RNA labeling-based scRNA-seq in answering fundamental questions about ZGA and cell cycle regulation in vertebrate development.

3. This manuscript is exclusively focused on the metabolic labeling as a result of chemical-induced conversion of bases such as using the nucleoside analog 4-SU, which is indeed powerful and has wide applications. However, there are many other metabolic labeling approaches to profile nascent RNA transcripts in single cells, which are not mentioned or discussed in depth by the authors. For example, recently, using the click chemistry-based approaches, Philip Sharp group developed scGRO-seq (Mahat et al., Nature 2024) to sequence the single-cell nascent RNAs, which revealed new mechanisms of coordinated transcription. While it is beyond the scope of the current work to experimentally compare the efficiency of detecting nascent transcripts metabolically labeled by differential modified bases in single cells, which is of course both technically and biologically important, the authors should at least cite those works and provide sufficient justification and discussion of why the chemical-induced conversion methods they focused on are superior than other metabolic labeling approaches - this would not only help contextualize this work but also magnify its significance.

Response 14 : We appreciate the reviewer's insightful comments and have revised the discussion (**Lines 402-412 in the revised manuscript**) to better contextualize our work by incorporating alternative metabolic labeling strategies, including scGRO-seq.

Our study focuses on 4sU labeling, which provides a high-throughput approach to track newly synthesized RNA and integrates well with droplet-based scRNA-seq platforms. However, we acknowledge that click chemistry-based methods, such as scGRO-seq¹⁶, offer complementary insights by directly capturing transcribing RNA polymerases, providing a precise view of transcriptional burst kinetics and enabling the detection of non-polyadenylated RNAs, such as replication-dependent histones.

While both 4sU labeling and scGRO-seq enable nascent RNA tracking, they differ in key aspects. 4sU labeling offers a broad perspective by capturing all newly synthesized RNAs, including processed transcripts, allowing for RNA turnover analysis. In contrast, scGRO-seq focuses on active transcription, providing real-time insights into transcription initiation and elongation but lacking information on RNA stability. Additionally, 4sU labeling is widely compatible with high-throughput scRNA-seq platforms, whereas scGRO-seq requires additional run-on and click chemistry steps, limiting its scalability.

Rather than being competing methods, combining 4sU labeling with scGRO-seq could provide a comprehensive view of RNA dynamics. While the compatibility of 4sU labeling with click chemistry requires further validation, a more practical approach would be to apply metabolic labeling scRNA-seq methods, such as scNT-seq, alongside scGRO-seq on the same samples and integrate the data computationally to capture both newly synthesized transcripts and actively transcribing RNA polymerases.

4. This study is specifically focused on profiling the poly(A)-tailed mRNA; however, as many non-coding RNAs, such as microRNAs, lincRNAs and repetitive endogenous retrovirus elements, that play essential roles in early embryogenesis, are activated during the zygotic genome activation. To generalize its application and draw solid conclusions, it would be important for the authors to extend this work to those non-mRNAs too as these scRNA-seq approach could be invaluable in revealing novel insights into gene regulation. We do not request the authors to perform additional experiments, but they should point out the limitations of this work somewhere in the manuscript.

Response 15 : We sincerely thank the reviewer for raising this important point. Our study focuses on poly(A)-tailed mRNA because current metabolic labeling-based scRNA-seq platforms, such as scNT-seq¹⁰, sci-fate^{1,2}, and Well-TEMP-seq⁴, rely on oligo(dT) priming, limiting their ability to capture non-polyadenylated RNAs. Many non-coding RNAs, including microRNAs like miR-430, long non-coding RNAs (lincRNAs), and endogenous retroviral elements, play essential roles during zygotic genome activation (ZGA) but are largely missed by poly(A)-enrichment strategies^{13,17,18}.

Several poly(A)-independent scRNA-seq methods have been developed that could complement our approach. Smart-seq-total¹⁹ enables the detection of non-coding RNAs by adding a poly(A) tail, snRandom-seq²⁰ captures all RNA types using random hexamer priming, and scGRO-seq¹⁶ directly detects nascent, non-polyadenylated transcripts, providing a real-time view of transcriptional activity. Although extending our study to non-coding RNAs is beyond its current scope, our optimized chemical conversion conditions could be adapted for these technologies in future studies. This would allow tracking non-coding RNA dynamics during early development, providing insights into miRNA-mediated mRNA clearance, repeat element activation, and lincRNA functions during ZGA.

We appreciate the reviewer's suggestions, which have improved the clarity and scope of our study. The relevant information has been incorporated into the discussion section of the revised manuscript (**Lines 394-412 in the revised manuscript**).

5. It is not surprising that the commercial MGI C4 outperforms other home-brew approaches, particularly when the cell numbers are rare which is often more than a concern for certain types of tissues such as early embryos. This conclusion overshadowed all other approaches presented by the authors, which is disappointing. That said, the hard work presented by the authors is still valuable, which serves a guidance for future method selection.

In the decision tree in Fig. 5, are there any other factors should be considered when choosing the methods and platforms (such as cost and cell types)?

Response 16 : We sincerely thank the reviewer for the insightful comments. We recognize that method selection should account for multiple factors beyond sequencing performance, including cost, cell type composition, and experimental feasibility. To address this, we have made the following improvements:

1) We expanded our benchmarking to include the widely used 10x Genomics platform alongside the commercial MGI C4 and home-brew Drop-seq platforms, systematically evaluating all possible combinations of chemical conversion methods and platforms (**Lines 281-294 in the revised manuscript**). Specifically, our results showed that on-bead chemical reactions consistently outperformed in-situ reactions, yielding higher T-to-C conversion rates and improved transcript recovery. However, 10x Genomics is incompatible with on-bead reactions, which limits its conversion efficiency. Therefore, we applied in-situ chemical conversion on 10x Genomics and compared its performance with on-bead conversion in MGI C4. At 4k reads per cell, library complexity was similar between 10x Genomics (nGene: 670, nUMI: 1,745) and MGI C4 (nGene: 624, nUMI: 1,263). However, within the same platform, on-bead conversion consistently outperformed in-situ conversion for both Drop-seq and MGI C4 (Revised **Fig. 4c-f**). These findings emphasize that method selection should consider not only sequencing performance but also the compatibility of chemical conversion techniques with different scRNA-seq platforms.

Revised Fig. 4c-f | Comparative analysis of single-cell platforms and chemical conversion methods. **c.** Proportion of UMIs containing T-to-C substitutions under different conditions and platforms. Different colored boxes represent various platforms and treatment methods. **d.** Box plot showing T-to-C substitution rates across the scRNA-seq platforms. **e-f.** Scatterplots showing the number of genes (e) or UMIs (f) detected per cell as a function of aligned reads per cell across the different platforms.

2) Based on the above results, we have incorporated cost per cell and features per platform into **Supplementary Table 3** and **Supplementary Fig. 11** (**Lines 295-304 in the revised manuscript**).

Supplementary Fig. 11

Revised Supplementary Fig. 11 | Decision Tree Summary of Chemical Conversion Methods and Platform Selection. The cost estimates are based on prices in mainland China, including additional expenses such as taxes, and may be subject to inflation.

Minor points:

1. Fig. 1 and Fig. 5 are helpful illustrations, but neither suffices a main figure. It is recommended to move them to the supplemental data.

Response 17 : We appreciate the reviewer's suggestion. We believe that Fig. 1 captures the core concept of the manuscript and is pivotal to its overall significance, making it essential to retain as a main figure. In response to your feedback, Fig. 5 have been enriched as follow and subsequently relocated to **Supplementary Fig. 11**.

2. Several figures (Fig 2c, 2d, 2f, 2g; Fig 4e, 4f) are hardly readable. The presentation of these figures should be improved by simplifying the plots.

Response 18: We apologize for any confusion caused. To enhance the clarity of the figures, we have 1) revised Fig. 2c,d by reordering the diagram in order from largest to smallest (Now **Fig. 2b,c**); 2) split the UMAP into 11 groups (Now **Supplementary Fig. 3b**); 3) summarized the numbers of genes and UMIs of each group in **Fig. 2b-c and Fig. 4e,f** and the labeled UMI proportion of each group in **Fig. 2g** into the Supplementary Table 1 and Supplementary Table 2.

Revised Fig. 2 | **b-c**. Scatterplots comparing the number of genes (c) or UMIs (d) detected per cell as a function of aligned reads per cell across the ten chemical conversion methods. Color indicates treatment methods.

Revised Supplementary Fig. 3b. Uniform Manifold Approximation and Projection visualization showing datasets respectively from control and the ten chemical conversion methods.

3. Negative controls and statistics for the ISH by each gene should be included in Fig. 3g.

Response 19 : We appreciate the reviewer's advice and have designed positive-sense probes for up to eight genes to serve as negative controls (**Revised Supplementary Fig. 8c**), as outlined in Response 10.

Revised Supplementary Fig. 8c. Antisense (top) and sense-strand negative control (NTC, bottom) across introns for whole-mount in-situ hybridization staining validation of 5.5 hpf zebrafish embryos for mRNAs of zygotic genes indicated in Fig. 3f,g. Scale bar: 200 µm.

4. It would be nice if the author could provide some rationale for using the ZF4 cell line for their "benchmark" analysis.

Response 20 : We thank the reviewer for highlighting these points. While 293T and K562 cell lines are more widely used, ZF4, a fibroblast cell line derived from zebrafish embryos, is better suited for our study as it aligns more closely with our focus on early zebrafish embryonic development.

5. For defining the cycling state, what data are used in Extended Data Fig. 2? More specific and detailed info how the authors processed them should be provided in the Methods and/or the legend.

Response 21 : We thank the reviewer for bringing this to our attention. We have included the additional details in the Methods section to ensure clarity and completeness (now **Supplementary Fig. 4; Lines 758-765 in the revised manuscript**).

6. The ambiguous expression should be clarified in Line 138: "This suggests that whether the chemical conversion is conducted in intact cells (in-situ) or after mRNA release (on-beads) introduces greater transcriptomic variation than the specific chemical conversion methods employed (Extended Data Fig. 2b)."

Response 22 : We apologize for any confusion caused. What we intended to convey was: “This indicates that performing chemical conversion in intact cells versus after mRNA release introduces more transcriptomic variation than the choice of chemical method alone” (now **Supplementary Fig. 4a; Lines 168-170 in the revised manuscript**).

7. References should be provided for the following:

(1) Line 39: “Metabolic RNA labeling combined with scRNA-seq has significantly enhanced our ability to quantitatively analyze RNA synthesis and degradation, leading to key discoveries.” Also, what “key discoveries” do the authors mean?

(2) Line 158: “Additionally, the top 10% most stable and unstable transcripts were enriched for GO terms similar to those found in mouse embryonic stem cells (Fig. 2h).”

(3) Line 268: “The results showed that mCPBA/TFEA-based methods maintained their efficiency, achieving T-to-C substitution rates comparable to, or slightly higher than, those observed in published datasets using fresh cells.”

Response 23 : We thank the reviewer for pointing these out.

1) The term “key discoveries” was explained in **Lines 46-49 in the revised manuscript** (marked red in the text).

2) We have provided appropriate references to support the statements highlighted (**Lines 188 and 349 in the revised manuscript**; marked red in the text).

8. The expression of “fair comparison” should be avoided due to its subjectivity. It should be “direct comparison”.

Response 24 : We appreciate the reviewer’s insightful suggestion and have incorporated the necessary revisions accordingly (**Lines 89, 209, 345 in the revised manuscript**; marked red in the text).

9. Some typos and grammatic errors should be corrected, including but not limited to:

(1) Line 42: “investigation of” should be “investigating”.

(2) Line 141: “Some maker genes” should be “Some marker genes”.

(3) Line 179: “and another” should be “and the other”.

(4) Line 289: “mutually” should be “multiple”.

Response 25 : We thank the reviewer for pointing out these issues. We have thoroughly reviewed the manuscript and address all typographical and grammatical errors to ensure clarity and accuracy:

1) **Line 48 in the revised manuscript**; 2) **Line 171 in the revised manuscript**; 3) **Line 208 in the revised manuscript**; 4) we have revised the related content (**Lines 368-372 in the revised manuscript**).

References:

- 1 Cao, J., Zhou, W., Steemers, F., Trapnell, C. & Shendure, J. Sci-fate characterizes the dynamics of gene expression in single cells. *Nat Biotechnol* **38**, 980-988, doi:10.1038/s41587-020-0480-9 (2020).
- 2 Maizels, R. J., Snell, D. M. & Briscoe, J. Reconstructing developmental trajectories using latent dynamical systems and time-resolved transcriptomics. *Cell Syst* **15**, 411-424 e419, doi:10.1016/j.cels.2024.04.004 (2024).
- 3 Junyue Cao *et al.* Comprehensive single-cell transcriptional profiling of a multicellular organism. *Science* **357**, 661–667 (2017).
- 4 Lin, S. *et al.* Well-TEMP-seq as a microwell-based strategy for massively parallel profiling of single-cell temporal RNA dynamics. *Nat Commun* **14**, 1272, doi:10.1038/s41467-023-36902-5 (2023).
- 5 Yin, K. *et al.* Dyna-vivo-seq unveils cellular RNA dynamics during acute kidney injury via in vivo metabolic RNA labeling-based scRNA-seq. *Nat Commun* **15**, 9866, doi:10.1038/s41467-024-54202-4 (2024).
- 6 Yin, K. *et al.* Well-Paired-Seq: A Size-Exclusion and Locally Quasi-Static Hydrodynamic Microwell Chip for Single-Cell RNA-Seq. *Small Methods* **6**, e2200341, doi:10.1002/smt.202200341 (2022).
- 7 Macosko, E. Z. *et al.* Highly Parallel Genome-wide Expression Profiling of Individual Cells Using Nanoliter Droplets. *Cell* **161**, 1202-1214, doi:10.1016/j.cell.2015.05.002 (2015).
- 8 Zheng, G. X. *et al.* Massively parallel digital transcriptional profiling of single cells. *Nat Commun* **8**, 14049, doi:10.1038/ncomms14049 (2017).
- 9 Holler, K. *et al.* Spatio-temporal mRNA tracking in the early zebrafish embryo. *Nat Commun* **12**, 3358, doi:10.1038/s41467-021-23834-1 (2021).
- 10 Qiu, Q. *et al.* Massively parallel and time-resolved RNA sequencing in single cells with scNT-seq. *Nature Methods* **17**, 991-1001, doi:10.1038/s41592-020-0935-4 (2020).
- 11 Riml, C. *et al.* Osmium-Mediated Transformation of 4-Thiouridine to Cytidine as Key To Study RNA Dynamics by Sequencing. *Angew Chem Int Ed Engl* **56**, 13479-13483, doi:10.1002/anie.201707465 (2017).
- 12 Zimmer, J. T. *et al.* Improving the study of RNA dynamics through advances in RNA-seq with metabolic labeling and nucleotide-recoding chemistry. *bioRxiv*, doi:10.1101/2023.05.24.542133 (2023).
- 13 Lee, M. T. *et al.* Nanog, Pou5f1 and SoxB1 activate zygotic gene expression during the maternal-to-zygotic transition. *Nature* **503**, 360-364, doi:10.1038/nature12632 (2013).
- 14 Chen, H. & Good, M. C. Nascent transcriptome reveals orchestration of zygotic genome activation in early embryogenesis. *Curr Biol* **32**, 4314-4324 e4317, doi:10.1016/j.cub.2022.07.078 (2022).
- 15 Chan, S. H. *et al.* Brd4 and P300 Confer Transcriptional Competency during Zygotic Genome Activation. *Dev Cell* **49**, 867-881 e868, doi:10.1016/j.devcel.2019.05.037 (2019).
- 16 Mahat, D. B. *et al.* Single-cell nascent RNA sequencing unveils coordinated global transcription. *Nature* **631**, 216-223, doi:10.1038/s41586-024-07517-7 (2024).
- 17 Giraldez, A. J. *et al.* Zebrafish MiR-430 promotes deadenylation and clearance of maternal mRNAs. *Science* **312**, 75-79, doi:10.1126/science.1122689 (2006).
- 18 Ulitsky, I., Shkumatava, A., Jan, C. H., Sive, H. & Bartel, D. P. Conserved function of lincRNAs in vertebrate embryonic development despite rapid sequence evolution. *Cell* **147**, 1537-1550, doi:10.1016/j.cell.2011.11.055 (2011).
- 19 Isakova, A., Neff, N. & Quake, S. R. Single-cell quantification of a broad RNA spectrum reveals unique noncoding patterns associated with cell types and states. *Proc Natl Acad Sci U S A* **118**, doi:10.1073/pnas.2113568118 (2021).
- 20 Xu, Z. *et al.* High-throughput single nucleus total RNA sequencing of formalin-fixed paraffin-embedded tissues by snRandom-seq. *Nat Commun* **14**, 2734, doi:10.1038/s41467-023-38409-5 (2023).